# Asymmetric apical domain states of mitochondrial Hsp60 coordinate substrate engagement and chaperonin assembly

Julian R. Braxton [1,2,3], Hao Shao[2], Eric Tse [2], Jason E. Gestwicki [2]✉ & Daniel R. Southworth [2]✉

The mitochondrial chaperonin, mitochondrial heat shock protein 60 (mtHsp60), promotes the folding of newly imported and transiently misfolded proteins in the mitochondrial matrix, assisted by its co-chaperone mtHsp10. Despite its essential role in mitochondrial proteostasis, structural insights into how this chaperonin progresses through its ATP-dependent client folding cycle are not clear. Here, we determined cryo-EM structures of a hyperstable disease-associated human mtHsp60 mutant, V72I. Client density is identified in three distinct states, revealing interactions with the mtHsp60 apical domains and C termini that coordinate client positioning in the folding chamber. We further identify an asymmetric arrangement of the apical domains in the ATP state, in which an alternating up/down configuration positions interaction surfaces for simultaneous recruitment of mtHsp10 and client retention. Client is then fully encapsulated in mtHsp60–10, revealing prominent contacts at two discrete sites that potentially support maturation. These results identify distinct roles for the apical domains in coordinating client capture and progression through the chaperone cycle, supporting a conserved mechanism of group I chaperonin function.

Many proteins require the assistance of molecular chaperones to assume their native conformation(s) in the cell[1]. Chaperonins are a highly conserved class of molecular chaperones found in all domains of life that form distinct multimeric ring complexes featuring a central cavity in which client protein substrates are folded[2,3]. Group I chaperonins, including the well-studied bacterial GroEL, form heptameric rings and require a co-chaperonin (here GroES) to cap the folding chamber and allow folding to occur. Group I chaperonin protomers are comprised of an apical domain (AD), an intermediate domain and an equatorial ATPase domain that coordinates inter-ring contacts to form a double-ring tetradecamer. Studies of bacterial GroEL–ES[4] identified that unfolded clients bind tightly to exposed, inward-facing

hydrophobic surfaces of the ADs and C-terminal tails found at the base of the folding cavity[5–8]. ATP binding results in a decreased affinity for client and an increased affinity for GroES[9–12]; GroES binding seals the now-hydrophilic cavity, favoring folding of the client protein[13–16]. GroES dissociates in a post-hydrolysis state, enabling the client to be released in a folded, native state or partially folded intermediate that requires subsequent rounds of chaperone interaction[17,18].

mtHsp60 is the only group I chaperonin found in humans. Along with its co-chaperonin, mtHsp10, it promotes the folding of proteins newly imported into the mitochondrial matrix, as well as proteins that have become denatured upon thermal or chemical stress[19–21]. This chaperonin has been implicated in the progression of several cancers[22,23],

[1]Graduate Program in Chemistry and Chemical Biology, University of California San Francisco, San Francisco, CA, USA. [2]Institute for Neurodegenerative Diseases, University of California San Francisco, San Francisco, CA, USA. [3]Present address: Division of Biology and Biological Engineering, California Institute of Technology, Pasadena, CA, USA. ✉e-mail: jason.gestwicki@ucsf.edu; daniel.southworth@ucsf.edu

and point mutations in mtHsp60 lead to severe neurodegenerative diseases known as hereditary spastic paraplegias, which cause progressive muscle spasticity and lower limb weakness[24–28]. Because of these links to disease, there is interest in understanding the structure and function of mtHsp60–10, and in developing inhibitors as chemical probes or potential therapeutics[29–32].

Advances in understanding the mtHsp60 mechanism have been limited by the instability of mtHsp60 complexes in vitro[33], likely explaining the lack of reported client-bound mtHsp60 structures. Thus, it is not clear which regions of mtHsp60 might be involved in these interactions or how clients might impact the mtHsp60 structure. However, given homologies between mtHsp60 and GroEL, the general chaperone mechanisms are thought to be conserved[34]. Indeed, client-free structures of mtHsp60 and mtHsp60–10 complexes identify similar conformational states[35–37], but there are important differences in ring–ring assembly and inter-ring allostery in these two systems[35–38]. Moreover, how mtHsp60 and mtHsp10 coordinate to bind and fold clients during ATP hydrolysis is unclear. Notably, for both mtHsp60 and GroEL, client and co-chaperone appear to bind to the same region[7], namely the inward-facing hydrophobic AD helices H and I. Thus, it is unclear how co-chaperonin binding can occur with a bound client given these overlapping interactions.

Here, we sought to structurally characterize mtHsp60 to identify its progression through the nucleotide- and mtHsp10-dependent chaperone cycle. Focusing on the disease-associated V72I mutant with increased oligomeric stability[25,27,39], we determined cryo-EM structures of apo-mtHsp60, ATP-mtHsp60 and ATP-mtHsp60–10 complexes. Subclassification reveals density corresponding to a bound client in the chamber that appears coordinated by distinct interaction sites in the apical and equatorial domains in each state. Notably, for ATP-mtHsp60 the ADs alternate between a client-contacting 'down' conformation and an outward-facing 'up' conformation. From these results we propose a mechanism in which AD up/down positioning enables client retention within the folding cavity to occur simultaneously with mtHsp10 recruitment, followed by AD rearrangement to an extended, symmetric position upon complete mtHsp10 binding. Together, these results provide insight into how group I chaperonins engage clients during their ATP-driven co-chaperonin recruitment cycle.

## Results

### mtHsp60[V72I] is hyperstable and retains chaperone activity

To facilitate structural studies of mtHsp60, we focused on the previously identified mtHsp60[V72I] variant that is associated with hereditary spastic paraplegia SPG13 (refs. 25,28), and is reported to have increased oligomeric stability[39]. Residue V72 is located in the equatorial domain of mtHsp60 and packs into its hydrophobic core, but does not contact the ATP-binding pocket (Fig. 1a and Extended Data Fig. 1a). Importantly, the V72I mutation retains some client refolding activity in vitro[39], suggesting that general features of the mtHsp60 chaperone cycle are preserved.

We first investigated V72I heptamer stability using size-exclusion chromatography coupled to multiangle light scattering (SEC-MALS). Under these conditions, mtHsp60[V72I] appears primarily heptameric, whereas wild-type mtHsp60 largely dissociates into monomers (Fig. 1b). Incubation with ATP causes complete dissociation of wild type, whereas an appreciable fraction remains oligomeric for mtHsp60[V72I]. Next, we analyzed the ATP hydrolysis activity of V72I and here identify an increase in activity relative to wild type (~10 pmol ATP hydrolyzed per min for wild type and ~21 pmol ATP per min for V72I) (Extended Data Fig. 1b). This is likely because of the enhanced oligomerization of mtHsp60[V72I] and resulting hydrolysis cooperativity[40,41]. Addition of mtHsp10 further stimulates hydrolysis in both wild type and V72I, although wild-type activity is slightly higher than V72I at high mtHsp10 concentrations (~37 versus 31 pmol min$^{-1}$, respectively). Overall, we conclude that the V72I mutation increases heptamer stability without

substantially altering ATP hydrolysis. Next, we measured mtHsp60–10 substrate refolding using chemically denatured mitochondrial malate dehydrogenase (mtMDH), a native client, using an NADH absorbance assay[42]. We identify that mtHsp60[V72I] is slightly impaired in client refolding activity compared with wild type (initial velocity of mtMDH activity ~0.009 $A_{340}$ per min for wild type versus ~0.006 $A_{340}$ per min for V72I) (Fig. 1c), indicating that this mutant retains a substantial amount of chaperone activity. In summary, we show mtHsp60[V72I] dramatically increases heptamer stability with modest functional effects compared with wild type in vitro, supporting further structural characterization.

### apo-mtHsp60[V72I] heptamers coordinate a bound client

We first sought to determine cryo-EM structures of the nucleotide-free (apo) mtHsp60[V72I] heptamer. Reference-free two-dimensional (2D) class averages show top views with clear heptameric rings and apparent $C7$ symmetry, and side views with two bands of density likely corresponding to the equatorial and ADs (Fig. 1d and Extended Data Fig. 1c,d). Remarkably, in certain top view class averages an additional asymmetric density in the central cavity is observed that we hypothesize to be a bound protein client (Fig. 1d). Initial three-dimensional (3D) classification of mtHsp60[apo] particles reveals four prevalent mtHsp60 heptamer classes (Extended Data Fig. 1e). Classes 2 and 4 feature density in the mtHsp60 central cavity, consistent with the top view 2D averages. In total, ~39% of the particles selected from 3D classification contain this density. Given that mtHsp60 heptamers are reconstituted from purified monomers and no additional protein was carried through the purification (Extended Data Fig. 1f), we conclude that the extra density is likely partially folded mtHsp60 that is retained as a client in the chamber. Indeed, mtHsp60 has been shown to be required for its own assembly into oligomeric complexes[43]. Moreover, the increased oligomer stability and slowed client folding activity of V72I may favor client-bound states, making mtHsp60[V72I] suitable for determining the structural basis of mtHsp60 chaperone function. Below, 'mtHsp60' refers to the V72I mutant and wild-type is indicated where relevant.

Given the structural similarities between all mtHsp60 classes, we jointly refined particles from all four classes with $C7$ symmetry to improve resolution. This resulted in a consensus map at 3.4 Å resolution, which enabled building of an atomic model (Fig. 1e, Extended Data Fig. 2a and Table 1). All domains of mtHsp60 were modeled except the flexible C-terminal tails, which were not resolved. This model matches the structures of previously published mtHsp60 heptamers (Cα root mean square deviation (r.m.s.d.) of ~0.6–0.8 Å)[36,37]. Although the equatorial and intermediate domains are well-resolved in this map, density for the ADs, including the cavity-facing helices H and I, is weaker, indicating flexibility (Fig. 1f and Extended Data Fig. 1g). Additional density in the central cavity is observed only at very low thresholds, likely because of its heterogeneity. From this analysis, we wondered whether the weaker AD density was a result of independent motions of each protomer, or whether a series of discrete heptameric arrangements of these domains existed, possibly related to client binding.

To better resolve the ADs and potential client contacts we sorted mtHsp60 particles solely by AD conformation and client density, excluding signal from the relatively invariant equatorial and intermediate domains. Given the $C7$ symmetry of the mtHsp60 heptamer, this was achieved by focused classification using symmetry-expanded[44] particles to resolve symmetry-breaking conformations of ADs (Fig. 1g). This approach resulted in three types of classes: those with greatly improved and asymmetric AD density and weak to no client density (36 of 50 classes), those with strong density corresponding to client but more poorly resolved ADs (13 of 50) and a class that resembled the consensus reconstruction (with weak, symmetric AD density) (Extended Data Fig. 1e,h). The absence of strong client density in classes with well-resolved ADs is likely caused by AD signal driving the classification, rather than client. Inspection of a representative class with

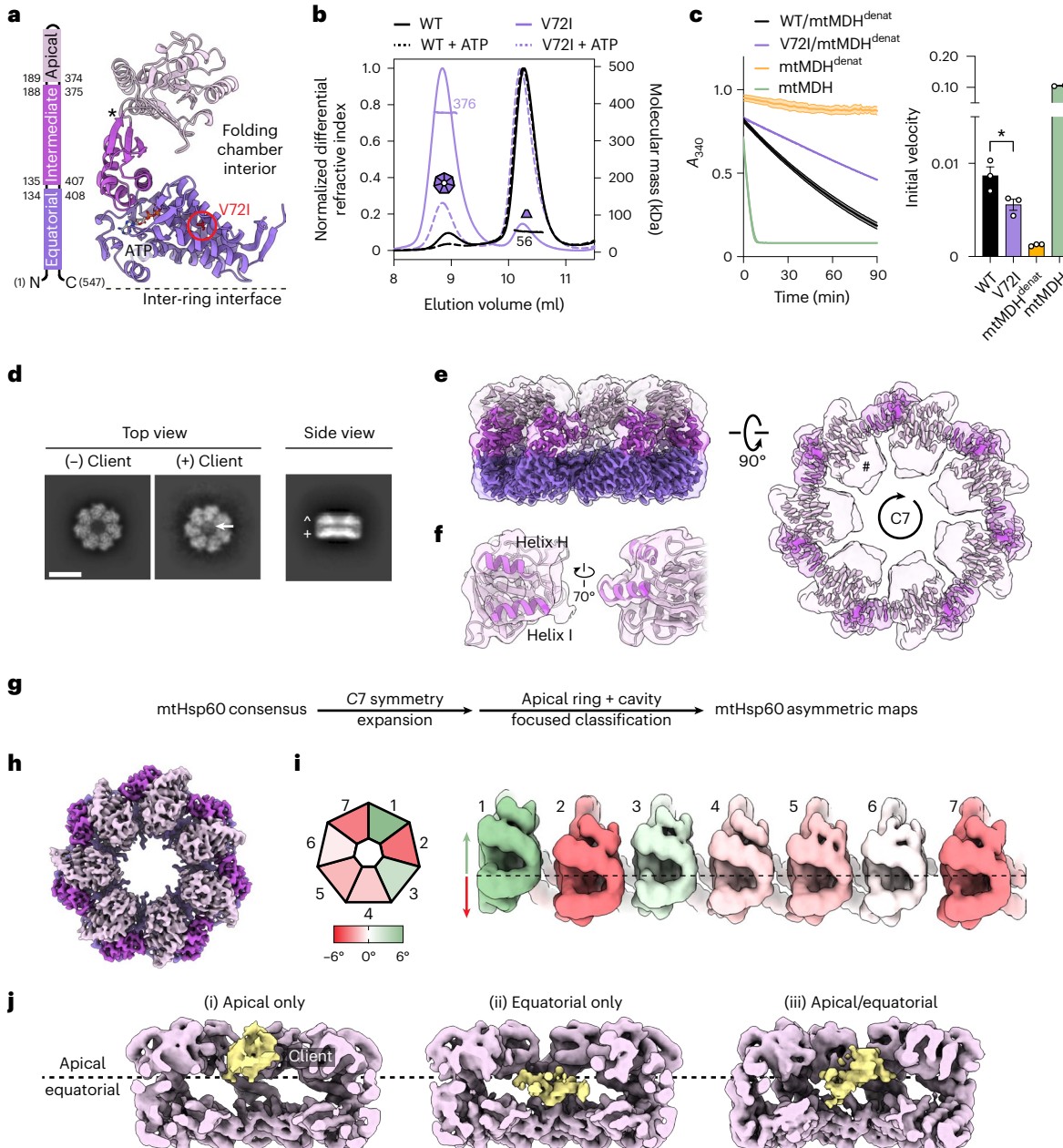

**Fig. 1 | Analysis and structures of apo-mtHsp60$^{V72I}$. a**, Structural schematic of mtHsp60 domains showing V72I mutation (red circle), ATP and the apical–intermediate domain hinge (asterisk). **b**, SEC-MALS of mtHsp60 (black) and mtHsp60$^{V72I}$ (purple) with (dashed lines) and without (solid lines) ATP. Normalized differential refractive index (left $y$ axis) and average molecular mass (horizontal lines, right $y$ axis) are shown versus elution volume ($x$ axis). This experiment was repeated a total of three times with similar results. **c**, mtHsp60–10 refolding of chemically denatured human mtMDH, measured by the decrease in NADH absorbance at 340 nm (left, one biological replicate, data are presented as the mean ± s.d. of technical triplicates). Initial velocities are shown (right, data are presented as the mean ± s.e.m. of two or three biological replicates, overlaid with individual values). Significance testing was performed by one-way analysis of variance, with correction for multiple comparisons by Dunnett's test. *$P$ = 0.0305. Folded (green) and denatured mtMDH (mtMDH$^{denat}$) (orange) are shown for comparison. **d**, Top and side view 2D class averages of client-bound and client-unbound mtHsp60$^{V72I}$ with client (arrow), and equatorial (+) and apical (^) domains indicated (Scale bar, 100 Å). **e**, Top and side views of the sharpened (opaque) and unsharpened (transparent) mtHsp60$^{apo}$ consensus maps are shown, colored as in **a**. AD density is absent in the sharpened map contour, indicated in the top view (#). **f**, Map and model view of an mtHsp60$^{apo}$ AD, with inward-facing helices H and I (dark purple) that are poorly resolved and flexible. **g**, Cryo-EM processing workflow to obtain maps with client and asymmetric AD conformations. **h**, Top view of the sharpened mtHsp60$^{apo}$ focus map, showing improved AD density. **i**, Heptamer schematic (left) and unwrapped view of the ADs of the unsharpened mtHsp60$^{apo}$ focus map colored to indicate AD rotation. Positive values (green) indicate an upward rotation (increased equatorial–apical distance) and negative values (red) indicate a downward rotation relative to the consensus map (dashed line). **j**, Slabbed side views of focused classifications showing unannotated client density (yellow) in different positions across the apical/equatorial regions.

improved AD density, termed mtHsp60$^{apo}$ focus, revealed a range of AD conformations around the heptamer, each related by a rigid-body rotation about the apical–intermediate domain hinge (Fig. 1h,i and Extended Data Fig. 2b). Relative to the consensus map, ADs rotate both

upward (away from the equatorial domain) and downward; the range of rotation among all protomers spans ~10° (Fig. 1h,i and Extended Data Fig. 1i). Intriguingly, some of the largest differences in AD position occur in adjacent protomers, giving rise to an apparent 'up' and 'down'

**Table 1 | Cryo-EM data collection, refinement and validation statistics**

| | mtHsp60$^{apo}$ consensus (EMD-29813, PDB 8G7J) | mtHsp60$^{apo}$ focus (EMD-29814, PDB 8G7K) | mtHsp60$^{ATP}$ consensus (EMD-29815, PDB 8G7L) | mtHsp60$^{ATP}$ focus (EMD-29816, PDB 8G7M) | mtHsp60$^{ATP}$–mtHsp10, consensus (EMD-29817, PDB 8G7N) | mtHsp60$^{ATP}$–mtHsp10 focus (EMD-29818, PDB 8G7O) |
|---|---|---|---|---|---|---|
| **Data collection and processing** | | | | | | |
| Microscope and camera | Glacios, K2 | Glacios, K2 | Glacios, K2 | Glacios, K2 | Glacios, K2 | Glacios, K2 |
| Magnification | 53,937 | 53,937 | 53,937 | 53,937 | 53,937 | 53,937 |
| Voltage (kV) | 200 | 200 | 200 | 200 | 200 | 200 |
| Data acquisition software | SerialEM | SerialEM | SerialEM | SerialEM | SerialEM | SerialEM |
| Exposure navigation | Image shift | Image shift | Image shift | Image shift | Image shift | Image shift |
| Electron exposure (e⁻/Å²) | 66 | 66 | 66 | 66 | 66 | 66 |
| Defocus range (μm) | −0.5 to −2.5 | −0.5 to −2.5 | −1.0 to −2.5 | −1.0 to −2.5 | −0.5 to −3.0 | −0.5 to −3.0 |
| Pixel size (Å) | 0.927 | 0.927 | 0.927 | 0.927 | 0.927 | 0.927 |
| Symmetry imposed | $C7$ | $C1$ | $D7$ | $C1$ | $D7$ | $C1$ |
| Initial particle images (no.) | 970,545 | 970,545 | 696,761 | 696,761 | 113,390 | 113,390 |
| Final particle images (no.) | 839,799 | 192,680 | 405,263 | 137,145 | 81,840 | 37,628 |
| Map resolution (Å) | 3.4 | 3.8 | 2.5 | 3.2 | 2.7 | 3.4 |
| FSC threshold | 0.143 | 0.143 | 0.143 | 0.143 | 0.143 | 0.143 |
| Map resolution range (Å) | 2.5–8 | 3–6 | 2–6 | 2.5–8 | 2–7 | 2.5–8 |
| **Refinement** | | | | | | |
| Initial model used (PDB code) | 7AZP | 7AZP | 6MRC | 6MRC | 6MRC | 6MRC |
| Model resolution (Å) | 3.6 | 4.1 | 2.7 | 3.4 | 2.8 | 3.6 |
| FSC threshold | 0.5 | 0.5 | 0.5 | 0.5 | 0.5 | 0.5 |
| Map sharpening $B$ factor (Å²) | −221.8 | −157.4 | −127.9 | −120.0 | −120.6 | −91.8 |
| Model composition | | | | | | |
| Non-hydrogen atoms | 27,461 | 27,461 | 55,524 | 26,334 | 66,122 | 33,047 |
| Protein residues | 3,682 | 3,682 | 7,378 | 3,505 | 8,778 | 4,389 |
| Ligands | 0 | 0 | 42 | 21 | 42 | 7 |
| $B$ factors (Å²) | | | | | | |
| Protein | 67.39 | 67.39 | 107.82 | 107.52 | 39.67 | 90.40 |
| Ligand | N/A | N/A | 23.99 | 44.80 | 11.75 | 61.26 |
| R.m.s.d. | | | | | | |
| Bond lengths (Å) | 0.015 | 0.014 | 0.008 | 0.004 | 0.004 | 0.003 |
| Bond angles (°) | 1.525 | 1.509 | 1.350 | 0.940 | 0.972 | 0.572 |
| **Validation** | | | | | | |
| MolProbity score | 0.83 | 0.87 | 0.91 | 1.25 | 1.04 | 1.24 |
| Clashscore | 1.16 | 1.41 | 1.63 | 4.88 | 2.56 | 3.99 |
| Poor rotamers (%) | 0.00 | 0.00 | 0.79 | 0.00 | 0.20 | 0.00 |
| Ramachandran plot | | | | | | |
| Favored (%) | 99.24 | 98.99 | 99.37 | 99.17 | 99.04 | 97.78 |
| Allowed (%) | 0.68 | 0.82 | 0.63 | 0.83 | 0.96 | 2.22 |
| Disallowed (%) | 0.08 | 0.19 | 0.00 | 0.00 | 0.00 | 0.00 |

N/A, not applicable.

alternating conformation (Fig. 1i, protomers 7 to 3). In summary, we successfully resolved the flexible mtHsp60$^{apo}$ ADs and identify that they adopt discrete up or down positions around the heptamer, rather than being randomly oriented.

Representative client-containing maps from focused classification feature client at multiple locations in the mtHsp60 heptamer (Fig. 1j and Extended Data Fig. 1e). Density corresponding to client is at low resolution compared with the mtHsp60 protomers, but this result is expected for a partially folded protein that likely populates multiple conformations; low-resolution client density has also been observed in GroEL structures[7,8,13,45]. In all structures, client is asymmetrically positioned in the central cavity and contacts multiple mtHsp60 protomers, which is consistent with the finding that group I chaperonins use multiple ADs to engage client, and with previous observations of asymmetric client density in GroEL tetradecamers[6–8]. However, there are notable differences in client localization between classes, with density positioned adjacent to the mtHsp60 AD, the equatorial domain or both (i, ii and iii, respectively, in Fig. 1j). In the apical-only class, client density

is proximal to helices H and I (Extended Data Fig. 1k,j), which contain multiple hydrophobic residues shown to be critical for the binding of unfolded proteins to GroEL[5], and also form the surface engaged by mtHsp10[35,38]. Likewise, in the equatorial-only class, client density is located deeper in the central cavity and appears to interact with the disordered C-terminal tails that project into this cavity (Extended Data Fig. 1l); in the apical/equatorial class, both contacts are observed. Notably, all client-bound classes also feature asymmetric AD conformations (Extended Data Fig. 1j), indicating that apical flexibility is a general feature of mtHsp60[apo] and is not limited to either client-bound or client-unbound complexes.

## ATP induces alternating mtHsp60 AD conformations

We next sought to characterize AD conformations and client positioning in the uncapped, ATP-bound state. ATP binding favors the formation of double-ring tetradecamers[46], and reference-free 2D class averages indeed reveal a double-ring arrangement for the majority of particles (Extended Data Fig. 3a). Notably, top view averages show clear density corresponding to client, whereas side views show the ADs are more poorly resolved compared with the equatorial and intermediate domains (Fig. 2a and Extended Data Figs. 1d and 3a). Given the symmetric features of the complex, we refined the double-ring complex with D7 symmetry (Fig. 2b and Extended Data Fig. 3b). The resulting map has an overall resolution of 2.5 Å, with the highest resolution in the equatorial and intermediate domains and greatly reduced resolution for the ADs (Extended Data Fig. 2c and Table 1). Client appears as a diffuse central density at approximately the level of the ADs, and is likely less visible because of the imposition of symmetry. We built an atomic model by docking a previous mtHsp60 model[35] into the density and performing an all-atom refinement (Extended Data Fig. 3c). ATP is clearly resolved in the nucleotide-binding pocket, indicating that this structure corresponds to a prehydrolysis state (Extended Data Fig. 3d). Compared with the apo state, nucleotide-binding induces a downward ~20° rigid-body rotation of the intermediate domain over the equatorial nucleotide-binding pocket, positioning the catalytic aspartate (D397) in proximity to the ATP γ-phosphate (Extended Data Fig. 3e). The inter-ring interface closely matches that of other nucleotide-bound mtHsp60 cryo-EM structures, with protomers arranged in a staggered 1:2 conformation and presenting two sites of interaction to the opposite ring[35]. At the left interface, residues in helix P form polar and hydrophobic interactions between rings, whereas no contacts are observed at the right interface (Extended Data Fig. 3f). The mtHsp60 inter-ring interface is greatly reduced compared with those in analogous GroEL complexes (~1,170 Å² buried in mtHsp60 compared with ~2,500 Å² in GroEL)[15,47,48], where contacts are made at both interfaces.

To improve resolution of the ADs and investigate their arrangement and client positioning, we performed focused classification of D7 symmetry-expanded particles, using a mask encompassing the ADs and central cavity of one heptamer (Fig. 2c and Extended Data Fig. 3b). Of 50 classes, 10 have greatly improved AD density for several protomers; the number of protomers per heptamer with improved density varies between three and six. Intriguingly, similar to the apo state, we identify an up/down arrangement in all ADs, but with improved resolution. Four of the ten classes (1–4) have six well-resolved ADs in this pattern, and the symmetry-breaking protomer (that is, the protomer between an up and down protomer) exhibits much weaker density, likely because of an inability to stably adopt either conformation.

Refinement of the best focused class with six well-resolved ADs (class 1, determined qualitatively) using a mask around the entire heptamer yielded the mtHsp60[ATP] focus map (Fig. 2d, Extended Data Fig. 2d and Table 1). This structure features substantially improved density for six ADs, whereas that of the symmetry-breaking protomer remained poorly resolved. Although the equatorial and intermediate domains are symmetric and identical to the consensus structure, the ADs adopt distinct 'up' and 'down' states in an alternating

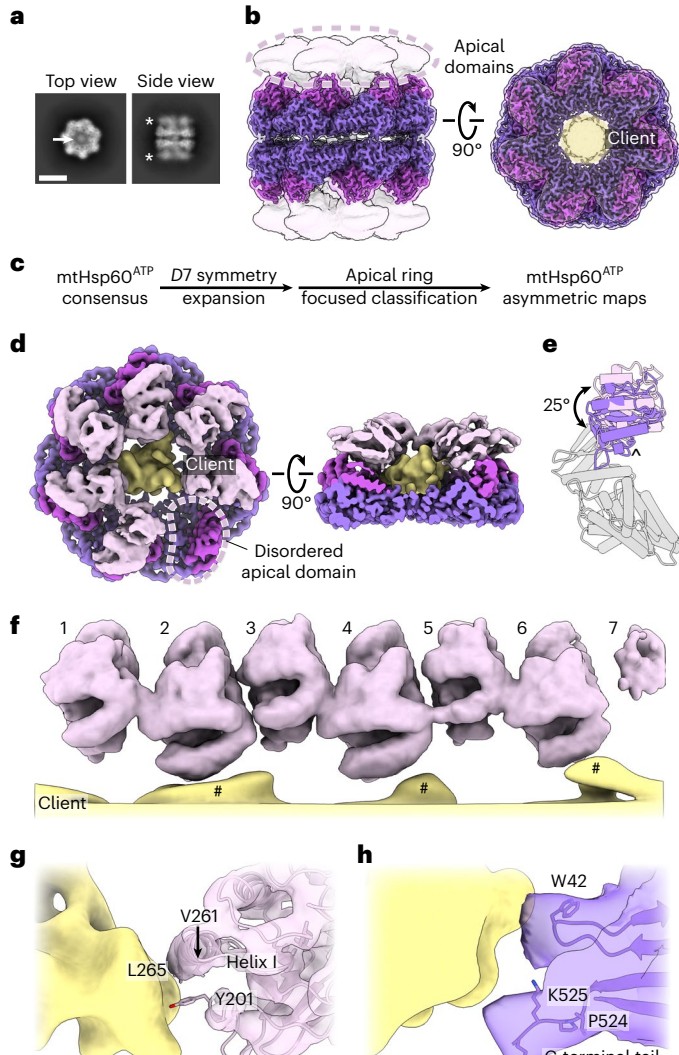

**Fig. 2 | ATP-mtHsp60[V72I] structure and client contacts. a**, Top and side view 2D class averages of ATP-bound mtHsp60[V72I]. Client density (arrow) and the flexible, lower resolution ADs (asterisk) are indicated. Scale bar, 100 Å. **b**, Sharpened (opaque) and unsharpened (transparent) maps of consensus ATP-bound mtHsp60[V72I], colored as in Fig. 1, showing lower-resolution AD density (circled) and the central density corresponding to client (yellow) in the top view (right). **c**, Cryo-EM processing workflow to capture asymmetric AD conformations. **d**, mtHsp60[ATP] focus map, shown as unsharpened mtHsp60 density overlaid with segmented and 8 Å low-pass filtered client density. Note the lack of density for one AD (circled). **e**, Aligned AD 'up' (pink) and 'down' (purple) conformations, showing a 25° rigid-body rotation between states. The AD underlying segment (below helices H and I) is indicated (^). **f**, Unwrapped view of the ADs and client in the mtHsp60[ATP] focus map from **d**, showing alternating up and down AD conformations. Note that client extensions (#) are only proximal to 'down' protomers (2, 4, 6), and the weak AD density for protomer 7 at the symmetry-mismatched interface. **g,h**, Putative 'down' AD (**g**) and equatorial domain (**h**) client contacts. Client is shown as an 8 Å low-pass filtered map and mtHsp60 is shown as models overlaid with transparent unsharpened maps.

arrangement around the heptamer. We identify that the up/down conformations are related by a rigid-body rotation of ~25° (Fig. 2e). The 'up' position displaces helices H and I from the central cavity; this likely eliminates potential client binding to these helices. By contrast, the 'down' position enables helices H and I to project directly into the central cavity. Apical inter-protomer contacts between 'up' and 'down' protomers are predominantly made using helices H and I, although the resolution is insufficient to identify specific interacting residues

(Extended Data Fig. 3g). Finally, modeling suggests that two adjacent 'up' protomers would not appreciably clash with each other but that two adjacent 'down' protomers would (Extended Data Fig. 3h). Given that adjacent 'up' protomers were not observed during focused classification it therefore appears that the alternating up/down arrangement is critical for stable AD positioning.

In addition to improved AD density, the mtHsp60$^{ATP}$ focus map features asymmetric client density in the mtHsp60 central cavity (Fig. 2d). As in mtHsp60$^{apo}$ structures, client is contacted by the apical and equatorial domains (Fig. 2f–h). Apical contacts are only made by 'down' protomers; this pattern of contact results in an asymmetric positioning in the mtHsp60 cavity (Fig. 2d). Based on our molecular model, these interactions primarily involve helix I and the underlying hydrophobic segment (Fig. 2g). The C-terminal tails and equatorial stem loop (residue W42) also contact client and, as in mtHsp60$^{apo}$, may serve to retain client in the folding cavity (Fig. 2h). This arrangement is distinct from the client densities identified in the apo state, likely because of the rotation of all ADs relative to those in apo states. In sum, we identify that ATP binding induces the alternating up/down conformational arrangement of ADs in all well-resolved classes. This causes asymmetry in the client-binding surface and potentially enables bifunctional interaction modes by the collective arrangement of ADs around the heptamer.

## mtHsp10 binding exposes distinct client-contacting surfaces

We next sought to determine structures of the mtHsp60–10 complex. We incubated these proteins with saturating ATP and prepared samples for cryo-EM as before. Reference-free 2D class averages reveal predominantly symmetric double-ring complexes (also referred to as 'footballs'), with a heptamer of mtHsp10 capping each mtHsp60 heptamer (Extended Data Fig. 4a,b). The structure of the football complex with *D*7 symmetry imposed refined to a resolution of 2.7 Å, with well-resolved density for mtHsp10 and all domains of mtHsp60, excluding the mtHsp60 C-terminal tails (Fig. 3a, Extended Data Fig. 2e and Table 1). In contrast to the apo and ATP structures, the ADs are well-resolved, similar to the equatorial and intermediate domains. Client is only observed in this consensus map at low thresholds, likely because of partial occupancy and heterogeneity in the central cavity relative to the strong density for mtHsp60 and mtHsp10.

To analyze the mtHsp10-bound state further, we built an atomic model (Extended Data Fig. 4c). ATP is well-resolved in the nucleotide-binding pocket and adopts the same orientation as in ATP-bound mtHsp60 (Fig. 3b). Likewise, the conformations of the equatorial and intermediate domains are almost identical to those in the ATP-bound state (Extended Data Fig. 4d). Relative to the 'up' ADs in mtHsp60$^{ATP}$, the ADs undergo an ~65° clockwise twist and elevation, generating a near-planar surface formed by helices H and I onto which mtHsp10 docks. mtHsp10 predominantly interacts with these helices through a hydrophobic triad (I31, M32, L33) in its mobile loop (Extended Data Fig. 4e). The interior of the mtHsp60–10 folding cavity features increased hydrophilicity relative to the interior of apo-mtHsp60 (Extended Data Fig. 4f), a pattern also observed in GroEL–ES complexes[49]. Finally, the inter-ring interface of this complex matches that of uncapped mtHsp60$^{ATP}$, indicating that no substantial changes in equatorial domain conformation are associated with mtHsp10 binding (Extended Data Fig. 4g).

To visualize client in the mtHsp10-bound state, we performed focused classification using a mask that included the folding cavity, with minimal density corresponding to mtHsp60 and mtHsp10 (Fig. 3c and Extended Data Fig. 4b). This approach resulted in a class with strong client density, which refined to 3.4 Å when using a mask encompassing the entire mtHsp60–10 ring (Fig. 3d). The bulk of the client density presents as a toroidal ring at approximately the level of the mtHsp60 ADs (Fig. 3d). mtHsp60–client contacts become apparent in low-pass filtered maps, revealing interactions with multiple mtHsp60 protomers

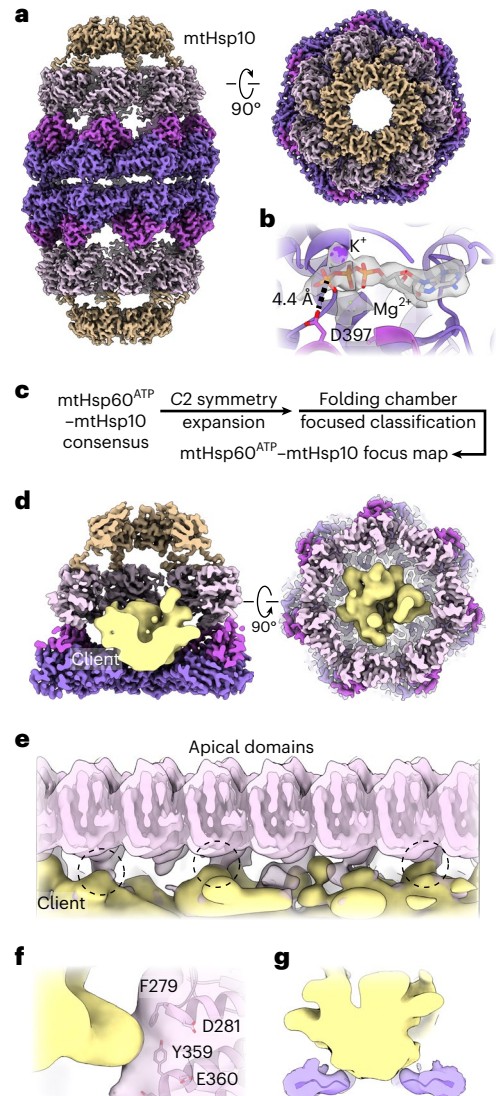

**Fig. 3 | ATP state mtHsp60–10 structure and client contacts. a**, Sharpened 2.7 Å resolution map of mtHsp60$^{ATP}$–mtHsp10, (mtHsp60 colored as in Fig. 1, mtHsp10 in brown). **b**, Nucleotide-binding pocket of mtHsp60$^{ATP}$–mtHsp10, showing density for ATP and the γ-phosphate thereof, and Mg$^{2+}$ and K$^+$ ions (gray, from sharpened map). **c**, Cryo-EM processing workflow to obtain the mtHsp60$^{ATP}$–mtHsp10 focus map with resolved client density. **d**, Slabbed views of the focus map showing the mtHsp10-capped chamber encapsulating client, shown as a segmented, 8 Å low-pass filtered density (yellow). **e**, Unwrapped view of the mtHsp60$^{ATP}$–mtHsp10 focus map from **d**, showing client contact with multiple ADs (circled). **f,g**, Enlarged map and model views of AD–client contacts with putative interaction residues labeled (**f**) and mtHsp60 C-terminal tail–client contacts (maps low-pass filtered to 8 Å) (**g**).

at the interface of two α-helical hairpins where aromatic residues F279 and Y359 project into the folding cavity (Fig. 3e,f). These residues are only exposed to the central cavity in the mtHsp10-bound state (Extended Data Fig. 4d). Contiguous density corresponding to client and the mtHsp60 C-terminal tails is also visible in filtered maps, suggesting that these extensions have a role during client folding (Fig. 3g). Overall, client localization and mtHsp60 contacts in this state resemble those in the mtHsp60$^{ATP}$ focus map and in client-bound GroEL–ES complexes[13,45], with both apical and equatorial domains in contact with client. This arrangement is distinct from the mtHsp60$^{apo}$ state, which features several client topologies, including apical-only and equatorial-only states, indicating a more heterogeneous association. Of note, multiple distinct conformations in the mtHsp10-bound complex

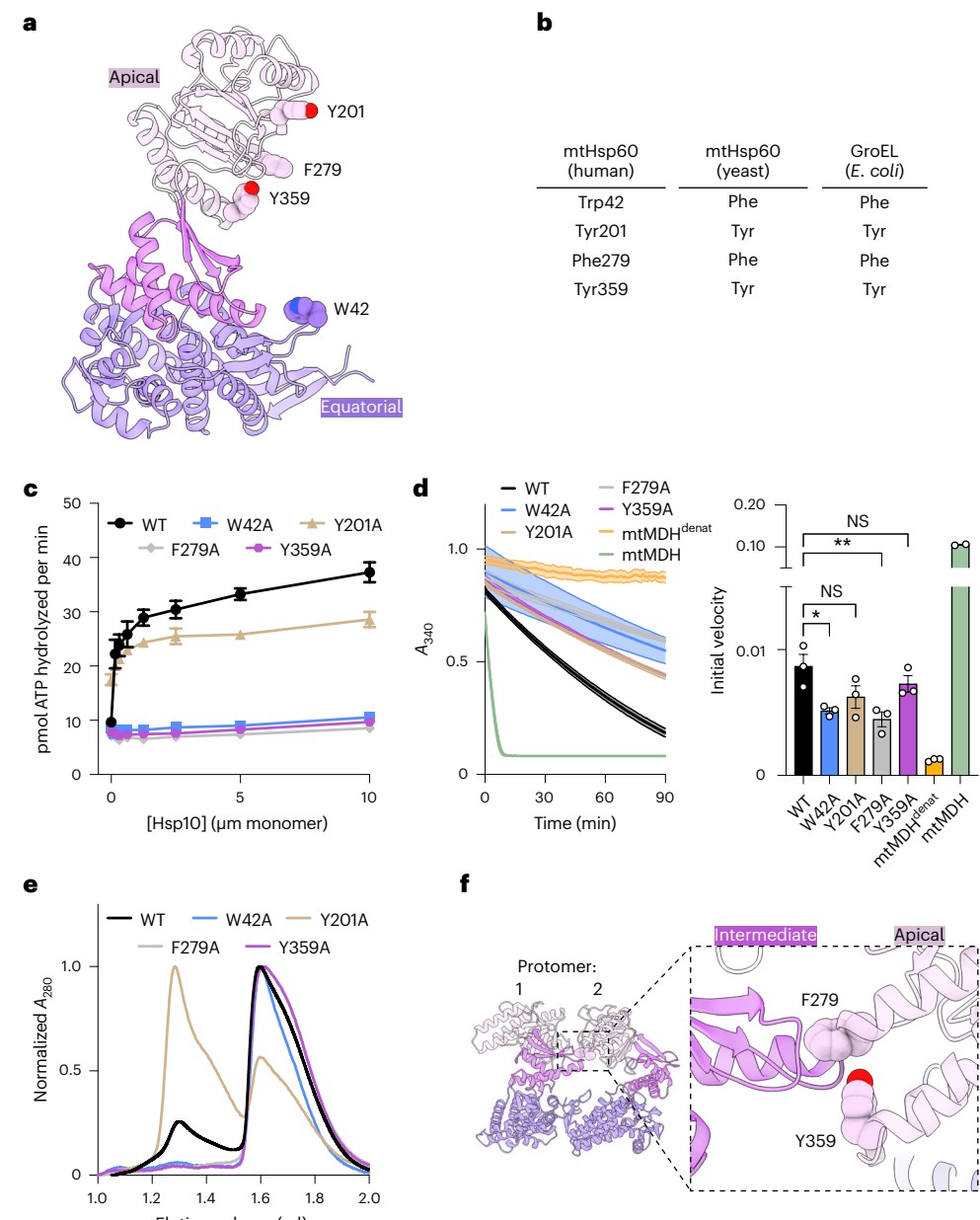

**Fig. 4 | Mutational analysis of client-contacting mtHsp60 residues.**
**a,b**, mtHsp60 protomer showing putative aromatic client-contacting residues
(**a**) and conservation across human and yeast mtHsp60 and GroEL (**b**). **c**, Steady-
state ATPase activity of mtHsp60 mutants versus concentration of mtHsp10.
A representative experiment of three biological replicates is shown; data are
presented as mean ± s.d. **d**, Enzymatic activity of chemically denatured human
mtMDH refolded by mtHsp60 mutants (left, representative of three biological
replicates). Data are presented as mean ± s.d. Initial velocities of absorbance
curves from two or three biological replicates are shown on the right; data are

presented as mean ± s.e.m. overlaid with individual values. Significance testing
was performed by one-way analysis of variance, with correction for multiple
comparisons by Dunnett's test. *$P$ = 0.0305; **$P$ = 0.0043; NS, not significant.
**e**, Analytical SEC traces of mtHsp60 mutants, showing complete monomerization
of W42A, F279A and Y359A mutants. This experiment was repeated a total
of three times with similar results. **f**, Model of two apo-mtHsp60 protomers,
showing AD residues F279 and Y359 contacting the intermediate domain of an
adjacent protomer.

might exist, although the likely heterogenous client population and
sub-stoichiometric occupancy probably precludes the identification
of distinct, or folded, conformations.

**Biochemical analysis of client-contacting mtHsp60 residues**
To probe the role of specific regions of mtHsp60 in client refolding
activity, we selected four residues observed to contact client and
mutated them to Ala in the wild-type mtHsp60 background to avoid
any contributing effects from the V72I mutation (Fig. 4a). W42 is located
on the equatorial domain stem loop, which is positionally invariant in

all mtHsp60 states. Y201 is located in the underlying segment of the
ADs, and is observed to contact client in the ATP state. F279 and Y359
contact client in the mtHsp10-bound state because of a substantial
rotation of the AD (Extended Data Fig. 4d); they do not face the fold-
ing chamber in the apo or ATP-bound states. Conservation analysis
between human mtHsp60 and its yeast and bacterial orthologs reveals
that three of these residues are conserved, whereas W42 is a Phe in the
other sequences (Fig. 4b). Analysis of ATPase activity reveals that the
activity of three of the four, W42A, F279A and Y359A, is not stimulated
by mtHsp10 (Fig. 4c). The activity of the Y201A mutant is modestly

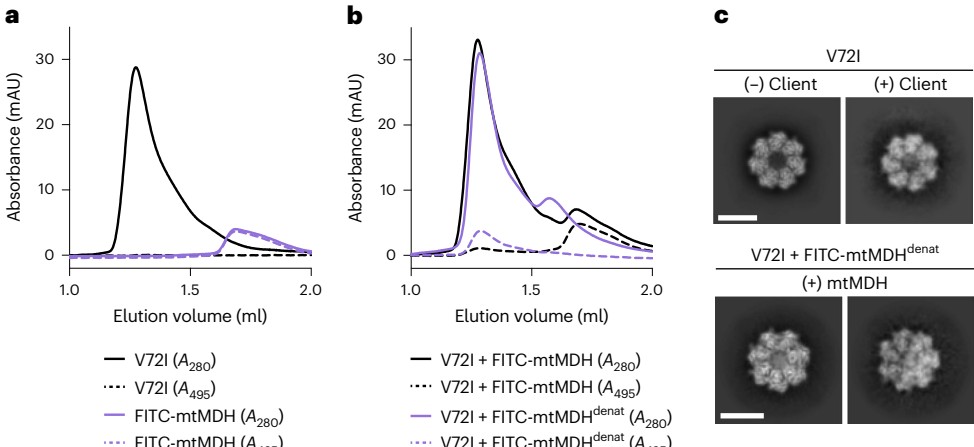

**Fig. 5 | Analysis of the mtHsp60–mtMDH client complex. a**, Analytical SEC traces of mtHsp60$^{V72I}$ (black) and folded FITC-mtMDH (purple). Solid lines are the $A_{280}$ traces and dashed lines are the $A_{495}$ traces. This experiment was repeated a total of three times with similar results. **b**, As in **a**, but for V72I incubated with folded FITC-mtMDH (black) or guanidine-denatured FITC-mtMDH (purple). Note the minimal coelution of mtHsp60 with folded FITC-mtMDH, compared with quantitative coelution with denatured FITC-mtMDH. This experiment was repeated a total of three times with similar results. **c**, Selected 2D top view class averages from the mtHsp60$^{apo}$ dataset (upper), showing classes with no or weak client density, and from the dataset with denatured FITC-mtMDH added (lower), showing much stronger client density. Scale bars, 100 Å. mAU, milli-absorbance unit.

impaired relative to wild-type mtHsp60 at high concentrations of mtHsp10, reminiscent of V72I (Extended Data Fig. 1b). Furthermore, all four mutants have impaired mtMDH refolding activity compared with wild type (Fig. 4d), a finding possibly explained by the perturbed ATPase activity. Given the lack of mtHsp10-stimulated ATPase activity in three of the four mutants, we next wondered whether these mutations had altered oligomerization propensities. Indeed, when analyzing these samples using size-exclusion chromatography (SEC) we observe that the W42A, F279A and Y359A mutants completely dissociate into monomers, whereas wild type and Y201A are at least partly heptameric (Fig. 4e). Inspection of the apo-mtHsp60 model reveals that F279 and Y359 are at an inter-protomer interface, and appear to contact the neighboring intermediate domain (Fig. 4f). Thus, mutation of these two residues potentially impairs this interaction, leading to a less stable heptamer. However, the mechanism of monomerization induced by the W42A mutation, which is not proximal to any inter-protomer interface, is less clear. We conclude that three of the four residues observed to contact client in the previous structures are also important for oligomerization. Thus, the investigation of mtHsp60 client refolding activity is confounded by an altered oligomeric state (monomer), given that the heptameric structure supports the folding of most (but not all[50]) clients.

### mtHsp60 interacts with and retains unfolded mtMDH
To further establish client interactions by mtHsp60, we next sought to complement our observations by analyzing mtHsp60$^{V72I}$ interactions with an established client, mtMDH. To that end, we fluorescently labeled recombinant mtMDH with fluorescein isothiocyanate (FITC-mtMDH) and analyzed its interaction with mtHsp60 using analytical SEC in both folded and denatured states (Fig. 5a,b). We specifically tracked FITC-mtMDH by measuring absorbance at 495 nm ($A_{495}$) (Fig. 5a). As shown in Fig. 1, mtHsp60$^{V72I}$ in isolation elutes predominantly as a heptamer (peak at -1.28 ml), whereas folded FITC-mtMDH elutes much later (peak at -1.68 ml), consistent with a dimeric state (Fig. 5a). When incubating mtHsp60 with folded FITC-mtMDH, the $A_{495}$ elution profile is essentially unchanged, indicating that this form is unable to efficiently interact with mtHsp60 heptamers (Fig. 5b). However, when mtHsp60 is incubated with chemically denatured FITC-mtMDH (FITC-mtMDH$^{denat}$), the $A_{495}$ peak shifts to an elution volume corresponding to that of mtHsp60 heptamers, indicating that mtHsp60

stably binds unfolded mtMDH. Notably, the $A_{280}$ peak that appears in this reaction (at -1.58 ml) likely corresponds to mtHsp60 heptamers that have dissociated into monomers during the incubation, because there is no corresponding $A_{495}$ signal (compare with the monomer peak in Fig. 4e). The enhanced mtHsp60 interaction with an unfolded client observed here is consistent with previous studies of client binding to chaperonins[51].

We next sought to correlate our biochemical analysis of the mtHsp60–FITC-mtMDH$^{denat}$ interaction with direct observation of these complexes using cryo-EM. 2D classification reveals several top view classes with density in the center of mtHsp60 heptamers, similar to but markedly more intense than that observed without the addition of mtMDH (Fig. 5c). This increased signal is likely a consequence of higher client occupancy. Of note, because of the lack of side view orientations in this dataset, we were unable to generate 3D reconstructions of these complexes. However, given the similar appearance of client density in mtHsp60 complexes with and without mtMDH, we conclude that the central densities in mtHsp60 structures determined here (Figs. 1–3) correspond to an unfolded client bound in the central cavity. Thus, the observed client positioning and interactions with mtHsp60 are likely authentic features of this chaperone.

### Model of mtHsp60 client engagement and chaperone cycling
The results presented here allow for the generation of a model describing client folding by the mtHsp60–10 system (Fig. 6 and Supplementary Video 1). In this model, mtHsp60 without nucleotide or co-chaperone exists as heptamers that are competent to bind client, with static equatorial and intermediate domains and somewhat flexible ADs loosely arranged in alternating up/down conformations. The client folding chamber in the apo state allows for multiple mtHsp60 interaction modes, including interaction with the inward surface of the ADs, the disordered C-terminal tails or both. ATP binding induces the dimerization of heptamers at the equatorial–equatorial interface, causes a downward rotation of the intermediate domain, closing the nucleotide-binding pocket and causing AD rotation. The ADs of ATP-bound protomers are arranged in a strict up/down alternating arrangement, with the 'down' protomers interacting with client through helix I and the underlying hydrophobic segment. Equatorial interactions, namely with the C-terminal tail and an aromatic residue projecting into the folding chamber, also contribute to client interaction. The 'up' ATP-bound

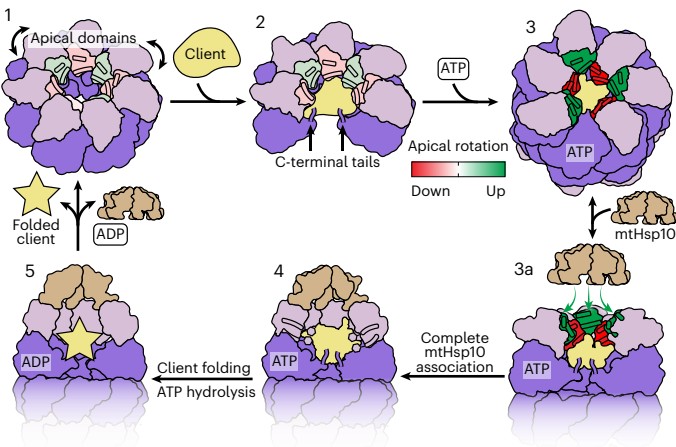

**Fig. 6 | Model of conformational changes in the client-engaged mtHsp60 reaction cycle.** State 1, ADs (pink) of mtHsp60[apo] heptamers are flexible and exhibit modest rotation about the apical–intermediate hinge, denoted by coloration of helices H and I. State 2, Client binding to mtHsp60[apo] preserves AD asymmetry, and client can localize to multiple depths of the heptamer, facilitated by mtHsp60 ADs and the flexible C-terminal tails. State 3, ATP binding induces the dimerization of heptamers through the equatorial domains and a more-pronounced AD asymmetry in an alternating up/down arrangement. ADs in 'down' protomers (red) contact client, whereas those in 'up' protomers (green) are competent to bind mtHsp10. State 3a, mtHsp10 initially binds the mtHsp60 heptamer using the three upward-facing ADs; all ADs then transition to the conformation observed in the mtHsp10-bound complex (state 4). After ATP hydrolysis and client folding (state 5), client, mtHsp10 and ADP are released, and the double-ring complex disassociates into heptamers.

protomers likely provide an initial platform for mtHsp10 association, and interaction with the remaining ADs induces the transition to a fully symmetric conformation that expands the now-capped folding chamber, allowing the client to fold. Finally, upon ATP hydrolysis mtHsp10 dissociates from the heptamer (as the affinity of ADP-bound mtHsp60 for mtHsp10 is negligible[52]), and client is released.

## Discussion

Chaperonins are a superfamily of molecular chaperones that promote protein folding by encapsulating unfolded or misfolded client proteins and allowing them to fold in a protected environment. How client and co-chaperonin binding are coordinated to enable efficient client folding in group I chaperonins, including the bacterial GroEL–ES and mito-chondrial Hsp60–10 systems, has remained an active area of study. The mtHsp60–10 system is a relatively understudied chaperonin homolog yet has critical roles in human health and disease. Here we used the stabilizing V72I mutant to structurally characterize intermediates in the mtHsp60 chaperone cycle. In each state, we observed low-resolution density in the mtHsp60 central cavity that corresponds to an unfolded client, likely unfolded mtHsp60 monomers. The assignment of this density as an authentic client is supported by investigations with mtMDH, which exhibits prototypical behavior with respect to mtHsp60 binding and is retained in a similar orientation in the mtHsp60 cavity (Fig. 5). These structural investigations, coupled with mutations and biochemical analysis, substantially increase our understanding of the mechanism of group I chaperonins.

Based on our structures we propose that AD asymmetry is a key feature of the mtHsp60 cycle. The mtHsp60 apo state may be initially encountered by client and features moderate AD flexibility, in agreement with other studies of mtHsp60 and its homologs[36,37,53]. We identify that ADs in intact apo heptamers exhibit loosely enforced alternating

arrangements of 'up' and 'down' ADs, rather than being randomly distributed (Fig. 1h,i). These arrangements do not appear to be induced by client binding, because classes without resolved client that exhibit these patterns were identified (Fig. 1i). It is therefore possible that these arrangements are simply more energetically favorable than in a perfectly symmetric AD ring, perhaps because of steric constraints. In addition, the multitude of AD configurations present in mtHsp60 complexes may enable the capture of a larger client repertoire than with a static heptameric arrangement because of differences in surface characteristics that may favor specific clients or folding states thereof, as discussed previously[54]. Intriguingly, similar AD arrangements are observed in ATP-bound structures, although the degree of asymmetry is greater and the up/down pattern is consistently observed across the different classes (Fig. 2f and Extended Data Fig. 3b). The positioning in the apo state may predispose the ADs for the alternating arrangement we observe in the ATP-bound state.

AD asymmetry may be a general feature of group I chaperonins, particularly given the high sequence identity between orthologs (Extended Data Fig. 5a). As with mtHsp60, most GroEL structures have been determined with symmetry applied, potentially obscuring conformational variability linked to function. However, one high-resolution GroEL structure[55] determined without symmetry in an uncapped, nucleotide-bound state features an AD arrangement intriguingly similar to that determined here, indicating that GroEL complexes may also feature substantial asymmetry in at least one state (Extended Data Fig. 5b,c). In this structure, the ADs (all of which are resolved) do not display the strict up/down pattern that defines mtHsp60[ATP] states, but rather appear randomly arranged in the heptamer. In addition, multiple AD inter-protomer contacts observed in other states are not present here, likely explaining the observed conformational plasticity. An additional cryo-EM study of GroEL shows similarly asymmetric states[56], further suggesting that asymmetry is a conserved feature of group I chaperonin structure.

AD asymmetry (particularly in the uncapped, ATP-bound state) may directly affect progression through the mtHsp60 chaperone cycle. The ATP dependence in mtHsp10 binding may be a consequence of the upward-positioned ADs, which appear optimally positioned to interact with the mtHsp10 mobile loop. Initial association of three mtHsp10 mobile loops with corresponding ADs (leaving the remaining four unbound and flexible) may be entropically favorable relative to a concerted and ordered binding of all seven domains. Of note, considerable intra-ring heterogeneity in transitions between apo, ATP-bound and GroES-bound GroEL complexes has been observed using molecular dynamics simulations[57], further establishing the importance of asymmetry in the chaperonin cycle.

How client proteins are retained in the folding chamber during co-chaperonin binding has remained an open question for all group I chaperonins. Here, the alternating AD arrangements observed in the ATP-bound states raise exciting hypotheses about how this objective is achieved. We speculate that ADs in the 'up' conformation enable efficient recruitment of co-chaperonin, whereas those in the 'down' conformation are poised to interact with client. This alternating arrangement would enable simultaneous client retention and co-chaperonin recruitment, likely preventing premature client release into solution during co-chaperonin association. The three 'up' ADs provide a platform for initial co-chaperonin association (Fig. 5, green apical surfaces), and subsequent conformational rearrangements propagated around the heptamer result in formation of the fully encapsulated complex. This model is consistent with previous biochemical studies of group I chaperonins, which suggest multiple ATP- and co-chaperonin-bound intermediates on the pathway to complete encapsulation[12,58]. Although additional states likely exist, this work defines distinct nucleotide-dependent structural intermediates of mtHsp60 that may enable simultaneous assembly of the active holo complex during its chaperone cycle.

## Online content

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

## Methods

### Molecular cloning

The coding sequences of mtHsp60 ('mature' construct, residues 27-end) and full-length mtHsp10 were cloned into pMCSG7, containing a TEV protease-cleavable N-terminal 6X His tag. The Q5 Site-Directed Mutagenesis kit (New England Biolabs) was used to introduce mutations into the mtHsp60 expression construct. The numbering of mtHsp60 residues corresponds to the mature mtHsp60 protein after cleavage of the mitochondrial import sequence.

### Protein expression and purification

Human mtHsp60 constructs and mtHsp10 were expressed and purified as previously described[33,42]. In brief, plasmids were transformed into *Escherichia coli* BL21(DE3) chemically competent cells (New England Biolabs) using standard protocols. BL21 cells were grown in Terrific Broth medium supplemented with 100 µg ml$^{-1}$ ampicillin at 37 °C with shaking until an optical density at 600 nm (OD$_{600}$) of ~1 was reached. Cultures were then induced with 400 µM IPTG and incubated at 37 °C for 4 h with shaking. Cells were harvested by centrifugation for 10 min at 4,000$g$, and stored at −80 °C until use.

All purification steps were performed at 4 °C unless otherwise specified. Cell pellets were resuspended in His-binding buffer (50 mM Tris pH 8.0, 10 mM imidazole, 500 mM NaCl), supplemented with EDTA-free protease inhibitor cocktail (Roche). The resuspensions were homogenized by douncing and lysed by sonication. Lysates were clarified by centrifugation at 35,000$g$ for 30 min. Lysate supernatants were incubated with HisPur Ni-NTA resin (Thermo Scientific) for 1 h. The resin was washed with His-washing buffer (50 mM Tris pH 8.0, 30 mM imidazole, 300 mM NaCl), and eluted with His-elution buffer (50 mM Tris pH 8.0, 300 mM imidazole, 300 mM NaCl). The 6X His tags were removed by incubating the eluates with TEV protease and 1 mM DTT for 4 h at room temperature, followed by overnight dialysis in SEC buffer (50 mM Tris pH 7.7, 300 mM NaCl, 10 mM MgCl$_2$). The next day, uncleaved protein was removed using a reverse nickel column and concentrated for reconstitution/SEC. mtHsp10 heptamers were purified on a HiLoad 16/600 Superdex 200 pg column (GE Healthcare) equilibrated in SEC buffer. mtHsp60 oligomers were reconstituted by mixing mtHsp60 with KCl, Mg(OAc)$_2$ and ATP in the following ratio: 573 µl of mtHsp60, 13 µl of 1 M KCl, 13 µl of 1 M Mg(OAc)$_2$ and 52 µl of 50 mM ATP. After incubation at 30 °C for 90 min, the mixture was applied to the same SEC column, and the oligomeric fractions were collected, supplemented with 5% glycerol, concentrated, flash frozen in liquid nitrogen and stored at −80 °C.

The mtMDH bacterial expression vector was a gift from N. Burgess-Brown (Addgene plasmid no. 38792; https://www.addgene.org/38792/). The vector was transformed into Rosetta 2(DE3)pLysS chemically competent cells (Novagen) using standard protocols. Rosetta 2 cells were grown in Terrific Broth supplemented with 50 µg ml$^{-1}$ kanamycin and 25 µg ml$^{-1}$ chloramphenicol at 37 °C with shaking until an OD$_{600}$ of ~1 was reached. Cultures were then induced with 500 µM IPTG and incubated at 18 °C overnight with shaking. Cells were harvested by centrifugation for 10 min at 4,000$g$ and stored at −80 °C until use.

All purification steps were performed at 4 °C. A cell pellet was resuspended in mtMDH His-binding buffer (50 mM HEPES pH 7.5, 20 mM imidazole, 500 mM NaCl, 5% glycerol), supplemented with EDTA-free protease inhibitor cocktail (Roche). The resuspension was homogenized by douncing and lysed by sonication. The lysate was clarified by centrifugation at 35,000$g$ for 30 min, filtered and applied to a 5-ml HisTrap column (GE Healthcare). The column was washed with 5 column volumes of mtMDH His-binding buffer, and eluted with a 10 column volume gradient of mtMDH His-elution buffer (50 mM HEPES pH 7.5, 250 mM imidazole, 500 mM NaCl, 5% glycerol). Fractions containing mtMDH were concentrated and injected onto a HiLoad 16/600 Superdex 200 pg column (GE Healthcare) equilibrated in mtMDH

SEC buffer (10 mM HEPES pH 7.5, 500 mM NaCl, 5% glycerol, 0.5 mM tris(2-carboxyethyl)phosphine). Fractions enriched in mtMDH were concentrated, flash frozen in liquid nitrogen and stored at −80 °C.

The purity of all proteins was verified by SDS−PAGE and concentration was determined using the Pierce BCA Protein Assay Kit (Thermo Scientific).

### SEC-MALS and analytical SEC

For SEC-MALS, mtHsp60 samples (17 µM monomer) incubated with 1 mM ATP where applicable were injected onto an SEC column (Shodex Protein KW-804) equilibrated at room temperature in MALS buffer (20 mM HEPES pH 7.5, 100 mM KCl, 10 mM MgCl$_2$) connected to an in-line DAWN HELEOS multiangle light-scattering detector and an Optilab T-rEX differential refractive index detector (Wyatt Technology). The molecular masses of proteins were determined using the ASTRA V software package (Wyatt Technology). For analytical SEC, mtHsp60 samples (17 µM monomer) were injected onto a Superdex 200 Increase 3.2/300 column equilibrated at room temperature in MALS buffer. The volume of all prepared samples was 60 µl.

For SEC experiments with mtMDH, mtMDH was labeled with FITC (Thermo Fisher Scientific) for 1 h at room temperature, and unreacted FITC was removed by desalting using a 10DG column (Bio-Rad). FITC-mtMDH was used as is or was chemically denatured for 1 h at room temperature with mtMDH denaturant buffer (50 mM Tris pH 7.4, 6 M guanidine HCl, 10 mM DTT). mtHsp60 (35 µM monomer) was incubated with folded or denatured FITC-mtMDH (10 µM monomer, 2× molar excess over mtHsp60 heptamer) for 10 min at room temperature, and injected onto a Superdex 200 Increase 3.2/300 column equilibrated at room temperature in MALS buffer. The absorbance at 280 nm (protein) and 495 nm (FITC) was recorded. The volume of all prepared samples was 60 µl.

### BIOMOL Green ATPase assay

ATPase activity was measured in 96-well plates using an assay reported previously[59], with minor modifications. In brief, 500 nM mtHsp60 monomer (final) was incubated with a twofold dilution series of mtHsp10, starting at 10 µM monomer (final), in ATPase buffer (100 mM Tris pH 7.4, 20 mM KCl, 6 mM MgCl$_2$, 0.01% Triton X-100). ATP was added to 1 mM (final), and the reactions (25 µl total) were incubated for 1 h at 37 °C. After incubation, 80 µl of BIOMOL Green reagent (Enzo Life Sciences) was added to each well, immediately followed by 10 µl of 32% w/v sodium citrate, to limit the nonenzymatic hydrolysis of ATP. The reactions were mixed and incubated at 37 °C for 15 min, then $A_{620}$ was measured on a SpectraMax M5 (Molecular Devices). ATP hydrolysis (pmol ATP hydrolyzed per min) was quantified using a standard curve of sodium phosphate and the following equation:

$$\text{pmol ATP hydrolyzed per min} = \frac{A_{620} \times \text{reaction volume (µl)}}{\text{Slope of standard curve } (A_{620}/\text{µM phosphate}) \times \text{incubation time (min)}}$$

### mtMDH refolding assay

mtMDH activity after refolding by mtHsp60−10 was measured using a previously reported assay with minor modifications[42]. To prepare chemically denatured mtMDH (mtMDH$^{denat}$), mtMDH was incubated for 1 h at room temperature in mtMDH denaturant buffer. A binary complex of mtHsp60−mtMDH$^{denat}$ was prepared by adding mtMDH$^{denat}$ (120 nM final) to mtHsp60 (3.33 µM final) in mtMDH reaction buffer (50 mM Tris pH 7.4, 20 mM KCl, 10 mM MgCl$_2$, 1 mM DTT), and incubating for 10 min at room temperature. mtHsp10 (6.67 µM final) was added to this mixture, and 30-µl aliquots were transferred to 96-well plates in triplicate. Then 20 µl of ATP was added to each well (1 mM final), and the reactions were incubated at 37 °C for 1 h. After incubation, an equivalent amount of mtMDH or mtMDH$^{denat}$ was added to the plate as controls for mtMDH activity, and 10 µl of 500 mM EDTA pH 8.0

was added to all wells to quench mtHsp60-mediated refolding. Then 20 µl of mtMDH enzymatic reporter (2.4 mM NADH, 20 mM sodium mesoxalate dissolved in mtMDH reaction buffer, freshly prepared for each assay) was added to all wells, and $A_{340}$ was measured using a SpectraMax M5 (Molecular Devices) for 90 min at room temperature. Initial velocities of NADH oxidation were calculated using the following equation:

$$\text{Initial velocity} \left( \frac{A_{340}}{\text{min}} \right) = -1 \times \frac{A_{340_{t=3\text{min}}} - A_{340_{t=0\text{min}}}}{3 \text{ min}}$$

Significance testing for calculated initial velocities was performed using Dunnett's multiple comparison test, using mtHsp60$^{WT}$ as the control.

## SDS–PAGE analysis
mtHsp60$^{V72I}$ (10 µl of 5 µM monomer) was loaded on a 4–15% TGX gel (Bio-Rad), run for 30 min at 200 V, and stained using Coomassie Brilliant Blue R-250 (Bio-Rad).

## Cryo-electron microscopy sample preparation, data collection and image processing
For apo-mtHsp60 samples, 2.4 mg ml$^{-1}$ mtHsp60 was prepared in ATPase buffer (without detergent), supplemented with 0.1% $n$-octyl-beta-D-glucopyranoside (Alfa Aesar) to improve particle orientation distribution. For samples with ATP, 0.2–0.6 mg ml$^{-1}$ mtHsp60 was prepared in ATPase buffer (without detergent), supplemented with 1 mM ATP. For samples with mtHsp10, 6.3 mg ml$^{-1}$ mtHsp60 and 1.3 mg ml$^{-1}$ mtHsp10 were prepared in ATPase buffer, supplemented with 1 mM ATP and 0.1% amphipol A8-35 (Anatrace) to improve particle orientation distribution. Then 3 µl of each sample was applied to glow-discharged (PELCO easiGlow, 15 mA, 2 min) holey carbon grid (Quantifoil R1.2/1.3 on gold or copper 200 mesh support), blotted for 3 s with Whatman Grade 595 filter paper (GE Healthcare), and plunge frozen into liquid ethane cooled by liquid nitrogen using a Vitrobot (Thermo Fisher Scientific) operated at 4 or 22 °C and 100% humidity. Samples were imaged on a Glacios transmission electron microscope (Thermo Fisher Scientific) operated at 200 kV and equipped with a K2 Summit direct electron detector (Gatan). Videos were acquired with SerialEM[60] in super-resolution mode at a calibrated magnification of 53,937, corresponding to a physical pixel size of 0.927 Å. A nominal defocus range of −1.0 to −2.0 µm was used with a total exposure time of 10 s fractionated into 0.1-s frames for a total dose of 66 e$^-$/Å$^2$ at a dose rate of 6 e$^-$ per pixel per s. Videos were subsequently corrected for drift, dose-weighted and Fourier-cropped by a factor of 2 using MotionCor2 (ref. 61).

For the apo-mtHsp60 dataset, a total of 20,223 micrographs were collected and initially processed in cryoSPARC[62] (Extended Data Fig. 1e). After Patch CTF estimation, micrographs were manually curated to exclude those of poor quality, followed by blob- or template-based particle picking, particle extraction with a 352 pixel box, 2D classification and ab initio modeling in cryoSPARC. Particles selected from 2D analysis were subjected to an initial 3D classification in RELION[63]. Four classes resembled mtHsp60 heptamers, some of which contained density in the central cavity likely corresponding to a bound client. The particles from these four classes were jointly refined in cryoSPARC with $C7$ symmetry imposed. This resulted in the mtHsp60$^{apo}$ consensus map, which featured well-resolved equatorial and intermediate domains but very poor density for the ADs, indicating substantial conformational heterogeneity. To improve the resolution of the ADs and resolve client, particles from this refinement were Fourier-cropped by a factor of 2 (resulting in a box size of 176 pixels), symmetry-expanded in $C7$ and subjected to focused classification (50 classes, regularization parameter $T = 40$, 25 iterations) without image alignment in RELION (hereafter referred to as skip-align focused

classification), using a mask encompassing all ADs and the central cavity (Extended Data Fig. 1e, transparent yellow volume). Of note, using a mask that contained more (for example, the intermediate domains) or less mtHsp60 density resulted in lower quality reconstructions with respect to mtHsp60 or client; thus, including only the ADs appears optimal. Similarly, using lower values of the regularization parameter $T$ (for example, 10, 20) produced inferior reconstructions, suggesting that increased weight on the experimental images is necessary to better resolve these conformations. The above focused classification resulted in 36 classes with greatly improved ADs in asymmetric conformations (for example, class 1), 13 classes with moderate AD resolutions but strong density corresponding to client (for example, classes 2–4), and 1 class that resembled the symmetric consensus refinement and is likely composed of particles unable to be properly classified (Extended Data Fig. 1h). Particles from classes 1–4 were re-extracted to the original box size (352 pixels) and locally refined without symmetry in cryoSPARC using default parameters, resulting in the final maps. These refinements were performed with a mask that encompassed the entire complex (heptamer and central cavity). Of note, local (rather than global) refinement was performed such that the pose of each symmetry-expanded particle from focused classification was maintained, rather than allowing a global search that would result in multiple copies of the same particle having identical orientations (inappropriate data duplication).

For the ATP-mtHsp60 dataset, a total of 15,900 micrographs were collected over three different sessions, and initially processed as for apo-mtHsp60 (Extended Data Fig. 3b). Two classes (1 and 2) from initial 3D classification in RELION, both double-ring tetradecamers with weak central density corresponding to client, were jointly refined in cryoSPARC with $D7$ symmetry enforced, yielding the mtHsp60$^{ATP}$ consensus map. The equatorial and intermediate domains were well-resolved in this map, but density for the ADs was extremely poor, indicating substantial conformational flexibility. To better resolve the ADs, particles were Fourier-cropped to 200 pixels, symmetry-expanded in $D7$ and subjected to skip-align focused classification (50 classes, $T = 40$, 25 iterations) using a mask that encompassed the ADs and central cavity of one heptamer (Extended Data Fig. 3b, transparent yellow volume). This yielded ten classes with between three and six ordered ADs; of note, it is unclear whether classes with fewer than six ordered ADs represent authentic states or artifacts of classification. The remainder of the classes had no ordered ADs. Re-extraction to the original box size and asymmetric local refinement in cryoSPARC (default parameters) of the best class with six ordered ADs (class 1) using a mask that covered the classified heptamer and central cavity yielded the mtHsp60$^{ATP}$ focus map.

For the ATP-mtHsp60–10 dataset, a total of 7,460 micrographs were collected and initially processed as for apo-mtHsp60 (Extended Data Fig. 4b). Two classes (1 and 2) from an initial 3D classification in RELION, both resembling double-ring complexes with each ring bound by mtHsp10, were jointly refined in cryoSPARC with $D7$ symmetry imposed, resulting in the mtHsp60$^{ATP}$–mtHsp10 consensus map. Weak density in the central cavities prompted further analysis to classify rings with and without client density. To this end, a mask was created encompassing the folding chamber of one ring, with minimal density for mtHsp60 or mtHsp10 (Extended Data Fig. 4b, transparent cyan volume). A skip-align focused classification ($T = 40$, 25 iterations) into two classes was performed in RELION on $C2$ symmetry-expanded particles, which resulted in classes with and without client density. The class with client was locally refined without symmetry in cryoSPARC using a mask that encompassed the entire ring, resulting in the mtHsp60$^{ATP}$–mtHsp10 focus map.

For the mtHsp60-FITC-mtMDH dataset, FITC-mtMDH was chemically denatured for 1 h at room temperature in mtMDH denaturant buffer. mtHsp60 (42 µM monomer) was incubated with denatured FITC-mtMDH (12 µM monomer, 2× excess over mtHsp60 heptamer) for

10 min at room temperature (100 µl total volume), and then desalted using a PD SpinTrap G-25 column (Cytiva). The eluate was diluted by a factor of six in ATPase buffer (without detergent), and 3 µl was applied to glow-discharged (PELCO easiGlow, 15 mA, 2 min) holey carbon grid (Quantifoil R1.2/1.3 on gold 200 mesh support), blotted for 3 s with Whatman Grade 595 filter paper (GE Healthcare), and plunged frozen into liquid ethane cooled by liquid nitrogen using a Vitrobot (Thermo Fisher Scientific) operated at 4 °C and 100% humidity. The sample was imaged on a Titan Krios TEM (Thermo Fisher Scientific) operated at 300 kV and equipped with a BioQuantum K3 Imaging Filter (Gatan) using a 20 eV zero-loss energy slit (Gatan). Some 4,998 videos were acquired with SerialEM[60] in super-resolution mode at a calibrated magnification of ×59,952, corresponding to a physical pixel size of 0.834 Å. A nominal defocus range of −0.8 to −1.8 µm was used with a total exposure time of 2 s fractionated into 0.0255-s frames for a total dose of 43 $e^-$/Å$^2$ at a dose rate of 15 $e^-$ per pixel per s. Videos were subsequently corrected for drift, dose-weighted and Fourier-cropped by a factor of 2 using MotionCor2 (ref. [61]). Dose-weighted sums were initially processed as for apo-mtHsp60, and 1,723,287 particles were extracted and subjected to 2D classification into 150 classes. 2D analysis revealed exclusively top views, a subset of which had markedly improved density corresponding to client. Further processing was not performed because of the lack of orientations necessary for 3D reconstruction.

## Molecular modeling

For the mtHsp60$^{apo}$ consensus structure, a previously published model (PDB 7AZP) was docked and refined against the sharpened map using Rosetta Fast Torsion Relax. The V72I mutations were made using Coot[64]. This model was then refined against the sharpened mtHsp60$^{apo}$ focus map. For the mtHsp60$^{ATP}$ consensus structure, a chain from a previously published model (PDB 6MRC) was docked into an asymmetric unit of the unsharpened map, and the AD was rigid-body docked using Phenix Real Space Refine[65]. The V72I mutation and ligand modifications were then made in Coot, followed by generation of the complete tetradecamer in Phenix and refinement against the sharpened map using Phenix Real Space Refine. One heptamer from this model was docked into the sharpened mtHsp60$^{ATP}$ focus map, and refined using Phenix Real Space Refine. The disordered AD was omitted from the model because of extremely poor resolution. For the mtHsp60$^{ATP}$–mtHsp10 consensus structure, a protomer pair of mtHsp60–10 from a previously published model (PDB 6MRC) was docked into an asymmetric unit of the sharpened map. The V72I mutation and ligand modifications were then made in Coot, followed by generation of the complete complex in Phenix and refinement against the sharpened map using Phenix Real Space Refine. One ring of this model was then docked into the sharpened mtHsp60$^{ATP}$–mtHsp10 focus map, and refined using Phenix Real Space Refine. Coot, ISOLDE[66] and Phenix were used to finalize all models.

## Calculation of buried surface area

The 'measure buriedarea' command with default parameters in UCSF ChimeraX was used to measure the buried surface area of the mtHsp60$^{ATP}$ consensus inter-ring interface, as well as of three GroEL structures (PDB 1AON, PDB 1SVT and PDB 1KP8), the values of which were averaged. Solvent molecules, if present, were excluded from the calculations.

## Protein sequence alignments

Amino acid sequences were aligned in MUSCLE[67] and visualized in MView[68].

## Data analysis and figure preparation

Biochemical data was analyzed and plotted using Prism (GraphPad). Figures were prepared using Adobe Illustrator, UCSF Chimera and UCSF ChimeraX[69,70].

## Reporting summary

Further information on research design is available in the Nature Portfolio Reporting Summary linked to this article.

## Data availability

Cryo-EM densities have been deposited at the Electron Microscopy Data Bank (EMDB) under accession codes EMD-29813 (mtHsp60$^{apo}$ consensus), EMD-29814 (mtHsp60$^{apo}$ focus), EMD-29815 (mtHsp60$^{ATP}$ consensus), EMD-29816 (mtHsp60$^{ATP}$ focus), EMD-29817 (mtHsp60$^{ATP}$– mtHsp10 consensus) and EMD-29818 (mtHsp60$^{ATP}$–mtHsp10 focus). Atomic coordinates have been deposited at the Protein Data Bank (PDB) under accession codes PDB 8G7J (mtHsp60$^{apo}$ consensus), PDB 8G7K (mtHsp60$^{apo}$ focus), PDB 8G7L (mtHsp60$^{ATP}$ consensus), PDB 8G7M (mtHsp60$^{ATP}$ focus), PDB 8G7N (mtHsp60$^{ATP}$–mtHsp10 consensus) and PDB 8G7O (mtHsp60$^{ATP}$–mtHsp10 focus). Accession codes for additional models referenced in this study are PDB 7AZP (mtHsp60 apo), PDB 6MRC (mtHsp60$^{ADP}$–mtHsp10), PDB 4KI8 (GroEL R-ADP), PDB 4AAR (GroEL Rs2), PDB 1AON (GroEL$^{ADP}$–GroES), PDB 1SVT (GroEL$^{ADP-AlFx}$–GroES) and PDB 1KP8 (GroEL$^{ATP}$). Source data are provided with this paper.

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

## Acknowledgements

We thank A. Brilot for helpful advice regarding cryo-EM data processing. This work was supported by the National Institutes of Health, grants F31GM142279 (to J.R.B.), NS059690 (to J.E.G.) and R01GM138690 (to D.R.S.).

## Author contributions

J.R.B. cloned mtHsp60 mutants, expressed and purified proteins, performed biochemical and cryo-EM experiments, built models, developed figures, and wrote and edited the manuscript. H.S. expressed and purified proteins. E.T. operated electron microscopes and assisted with data collection. J.E.G. and D.R.S. designed and supervised the project and wrote and edited the manuscript.

## Competing interests

The authors declare no competing interests.

## Additional information

**Extended data** is available for this paper at https://doi.org/10.1038/s41594-024-01352-0.

**Correspondence and requests for materials** should be addressed to Jason E. Gestwicki or Daniel R. Southworth.

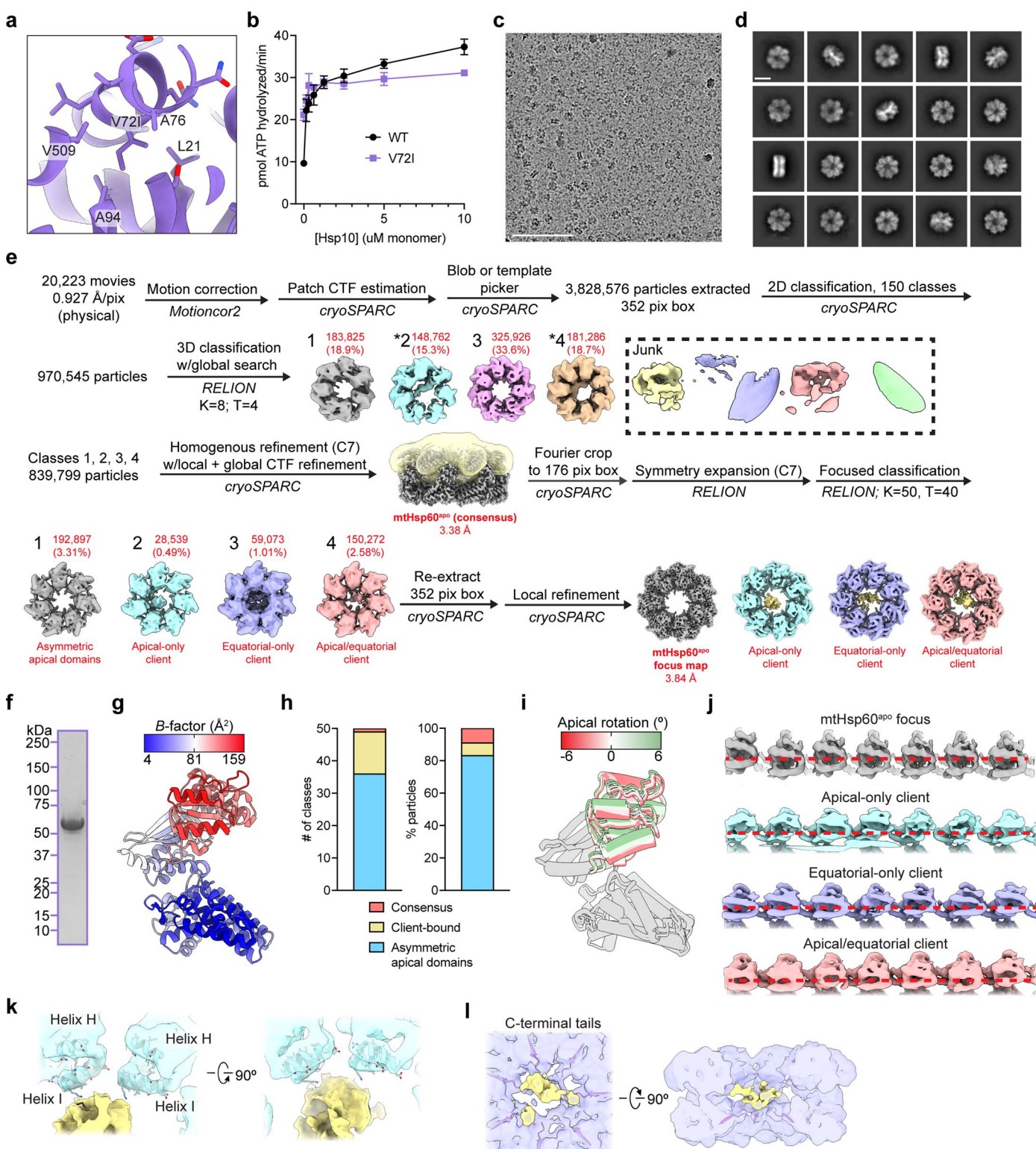

**Extended Data Fig. 1 | See next page for caption.**

**Extended Data Fig. 1 | Biochemical and cryo-EM analysis of apo mtHsp60$^{V72I}$.**
(**a**) View of V72I mutation in mtHsp60$^{apo}$, colored as in Fig. 1a. Adjacent
hydrophobic residues are also labeled. (**b**) Steady-state ATPase activity
of mtHsp60 (black) and mtHsp60$^{V72I}$ (purple) as a function of mtHsp10
concentration. A representative experiment of three biological replicates is
shown. Data are presented as mean ± s.d. (**c**) Representative micrograph from
the mtHsp60$^{apo}$ dataset. Scale bar equals 100 nm. (**d**) Representative 2D class
averages from the mtHsp60$^{apo}$ dataset. Scale bar equals 100 Å. (**e**) Cryo-EM
processing workflow for structures obtained from the mtHsp60$^{apo}$ dataset. The
mask used for focused classification is shown in transparent yellow with the
consensus map. Client-containing maps from the initial 3D classification are
indicated (*). (**f**) Coomassie Brilliant Blue-stained SDS-PAGE gel of recombinant
mtHsp60$^{V72I}$, showing no strong additional bands corresponding to other
proteins. This experiment was repeated a total of two times with similar

results. (**g**) Protomer of mtHsp60$^{apo}$ consensus colored by $B$-factor. (**h**) Class
distributions of the mtHsp60$^{apo}$ focused classification job, shown by number
of classes (left) and number of particles (right). Resulting classes either had
asymmetric apical domain conformations with improved map quality (blue,
relative to the consensus structure) or client density in the central cavity (yellow).
One class (salmon), representing ~9% of the data, resembles the consensus
state and likely represents particles that were not classified correctly (that is,
an artifact). (**i**) Overlay of mtHsp60$^{apo}$ focus protomers, with apical domains
colored as in Fig. 1i. (**j**) Unwrapped views of unsharpened mtHsp60$^{apo}$ focus and
client-bound maps, showing apical domain asymmetry and client density (where
applicable). Horizontal red dashed lines are for clarity. (**k**) Enlarged view of apical
domain helices H and I from the mtHsp60$^{apo}$ apical-only client map. (**l**) Enlarged
view of resolved portions of C-terminal tails from the mtHsp60$^{apo}$ equatorial-only
client map.

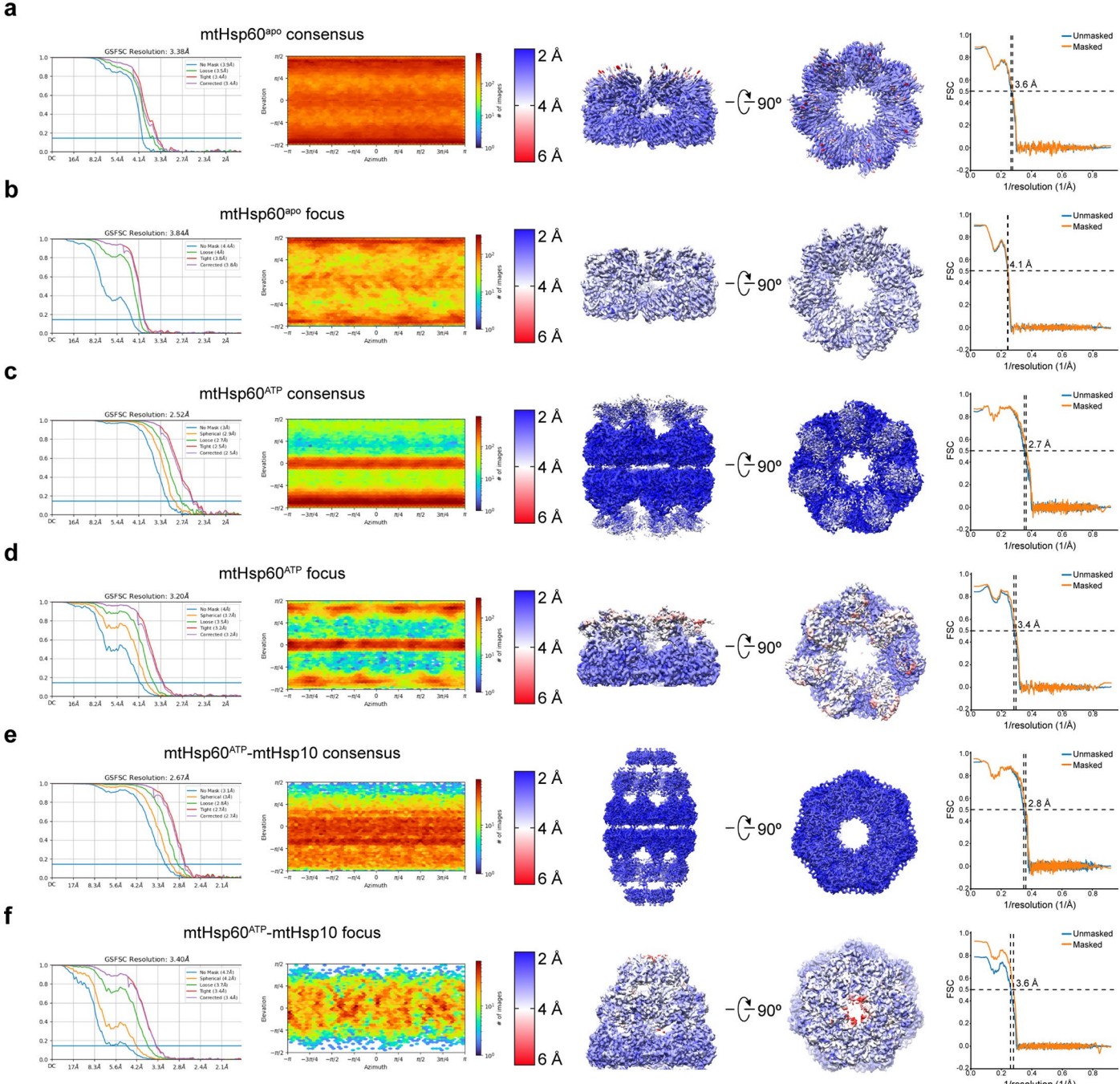

**Extended Data Fig. 2 | Cryo-EM densities and resolution estimation from the mtHsp60[V72I] datasets. (a to f)** Fourier Shell Correlation (FSC) curves, orientation distribution plots, sharpened maps colored by local resolution (0.143 cutoff), and map-model FSC curves for (**a**) mtHsp60[apo] consensus, (**b**) mtHsp60[apo] focus, (**c**) mtHsp60[ATP] consensus, (**d**) mtHsp60[ATP] focus, (**e**) mtHsp60[ATP]-mtHsp10 consensus, and (**f**) mtHsp60[ATP]-mtHsp10 focus structures. Displayed model resolutions for map-model FSC plots were determined using the masked map.

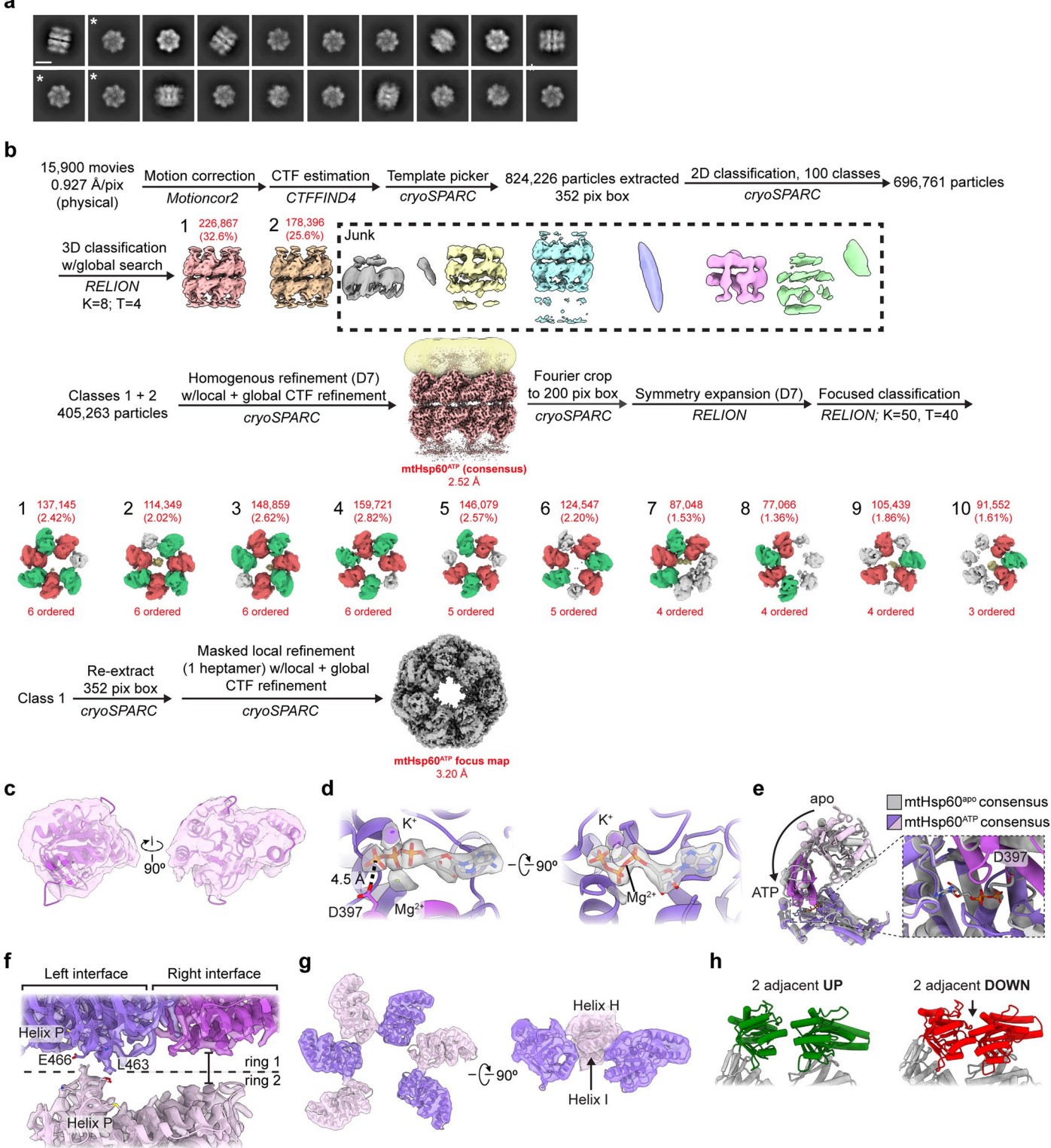

**Extended Data Fig. 3 | See next page for caption.**

**Extended Data Fig. 3 | Cryo-EM analysis of ATP-bound mtHsp60$^{V72I}$.**
(**a**) Representative 2D class averages from the mtHsp60$^{ATP}$ dataset. Scale bar equals 100 Å. Top views of single ring complexes are indicated (*). (**b**) Cryo-EM processing workflow for structures obtained from the mtHsp60$^{ATP}$ dataset. The mask used for focused classification is shown in transparent yellow with the consensus map. Protomers from focused classification maps are colored in green (apical domain facing upward), red (apical domain facing downward), or gray (disordered apical domain). Class 1 was selected for refinement based on visual assessment of map quality. (**c**) View of an apical domain from the unsharpened mtHsp60$^{ATP}$ consensus map and associated model. (**d**) Nucleotide binding pocket of mtHsp60$^{ATP}$, showing density for ATP and the γ-phosphate thereof, and Mg$^{2+}$ and K$^+$ ions (gray, from sharpened map). (**e**) Overlay of consensus

mtHsp60$^{apo}$ and mtHsp60$^{ATP}$ models, aligned by the equatorial domain, showing a downward rotation of the intermediate and apical domains in the ATP-bound state. (**f**) Inter-ring interface of the sharpened mtHsp60$^{ATP}$ consensus map and fitted model, showing contact at the left interface mediated by helix P, but no contact at the right interface. Each protomer is colored a different shade of purple. (**g**) Unsharpened map and model of ordered apical domains of mtHsp60$^{ATP}$ focus. 'Down' protomers are colored purple, 'up' protomers are colored pink. (**h**) Modeling of two adjacent ATP-bound 'up' (left) or 'down' (right) protomers, generated by aligning a copy of chain C of mtHsp60$^{ATP}$ focus with chain D (up pair) or a copy of chain D with chain C (down pair). A large clash is observed with two adjacent down protomers, while two adjacent up protomers appear compatible.

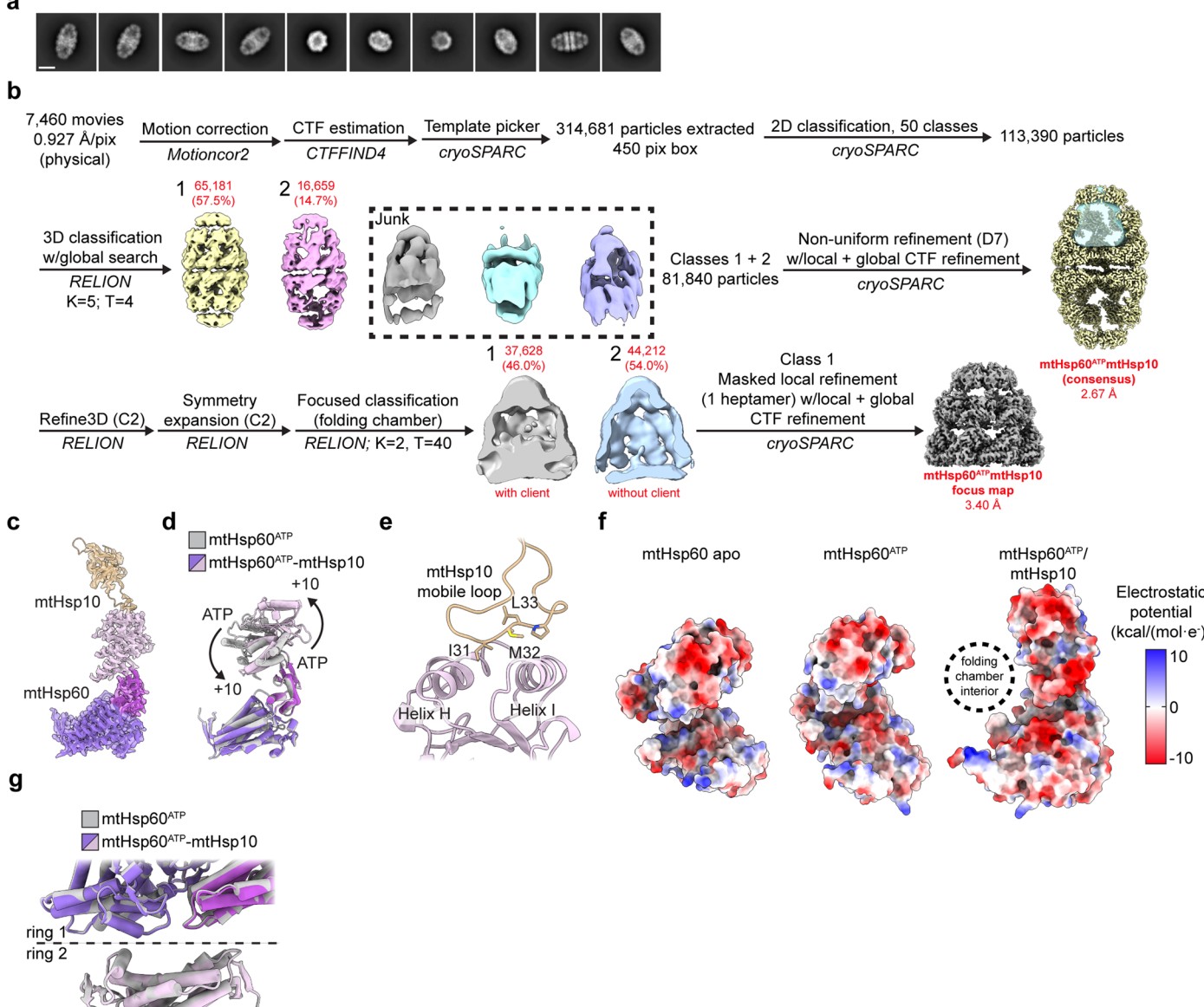

**Extended Data Fig. 4 | Cryo-EM analysis of ATP/mtHsp10-bound mtHsp60[V72I].**
(**a**) Representative 2D class averages from the mtHsp60[ATP]-mtHsp10 dataset. Scale bar equals 100 Å. (**b**) Cryo-EM processing workflow for structures obtained from the mtHsp60[ATP]-mtHsp10 dataset. The mask used for subsequent focused classification is shown (transparent blue) on the consensus D7 refinement. (**c**) Sharpened map and model for the asymmetric unit of the mtHsp60[ATP]-mtHsp10 consensus structure. (**d**) Overlay of consensus models for mtHsp60[ATP] and mtHsp60[ATP]-mtHsp10 structures, showing identical equatorial and intermediate domain conformations but a large upward apical domain rotation. (**e**) Model of the mtHsp10 mobile loop and associated mtHsp60 apical domain in the mtHsp60[ATP]-mtHsp10 consensus map, showing interaction of conserved hydrophobic residues with apical domain helices H and I. (**f**) Coulombic potential maps of protomers of mtHsp60 apo, mtHsp60[ATP], and mtHsp60[ATP]-mtHsp10 consensus structures, showing increased negative charge in the inward-facing regions of mtHsp60[ATP]-mtHsp10. (**g**) Overlay of consensus models for mtHsp60[ATP] and mtHsp60[ATP]-mtHsp10 structures, showing highly similar inter-ring conformations.

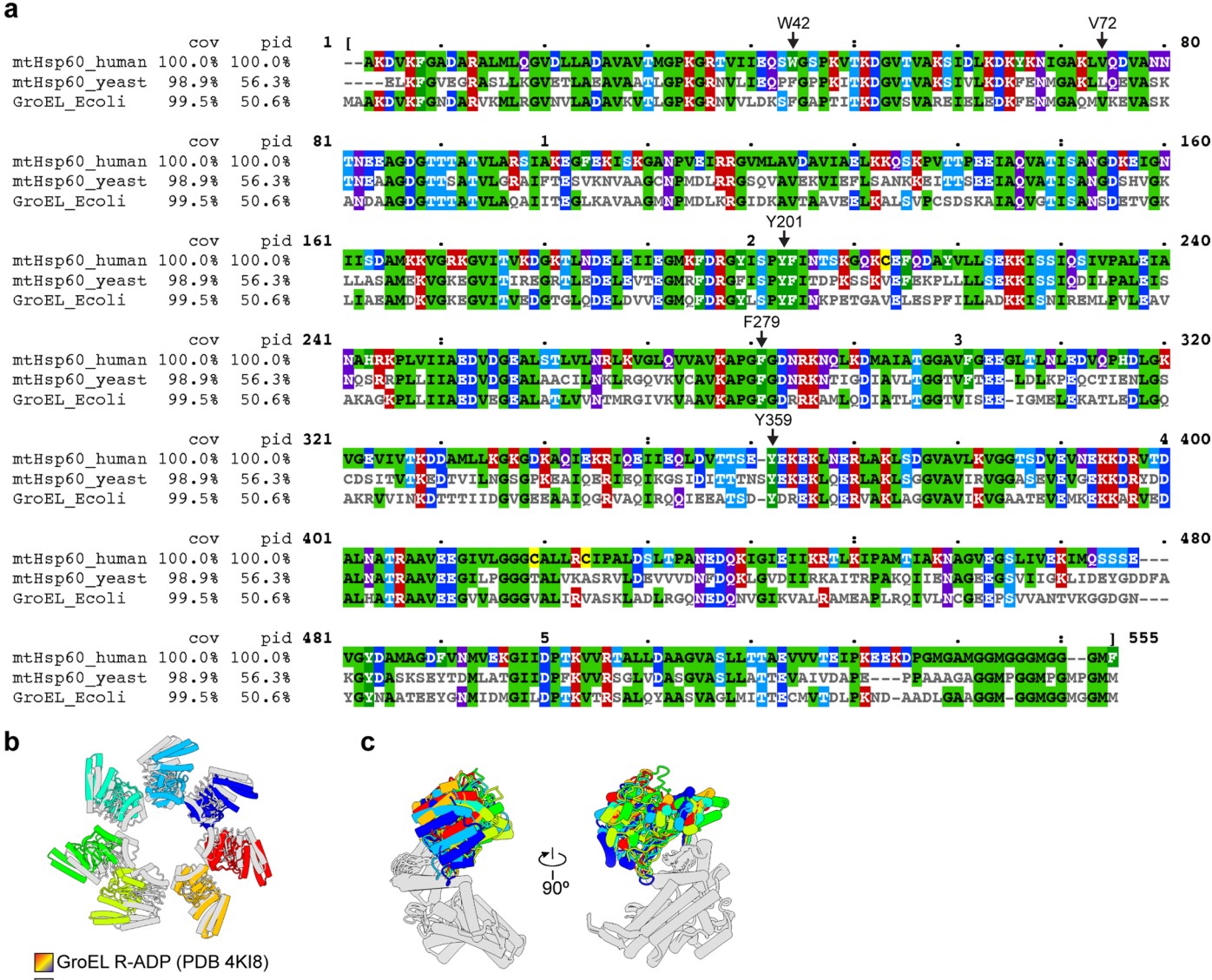

**Extended Data Fig. 5 | Sequence alignments and structural comparisons of group I chaperonins. (a)** Alignments of mature human (residues 27-end) and yeast (*Saccharomyces cerevisiae*, residues 26-end) mitochondrial Hsp60 and *E. coli* GroEL amino acid sequences. Residues mutated in this study are indicated (numbering corresponds to the human sequence). Cov = covariance relative to the human sequence, Pid = percent identity relative to the human sequence.

**(b)** Overlay of apical domains of asymmetric GroEL (R-ADP state, PDB 4KI8, colors) with symmetric state (Rs2, PDB 4AAR, gray), aligned by the equatorial and intermediate domains, showing deviations from C7 symmetry. **(c)** Overlay of all protomers of the GroEL R-ADP state (PDB 4KI8), aligned by the equatorial and intermediate domains, showing large variability in apical domain conformation. Apical domains are colored, other domains in gray.

# Reporting Summary

## Statistics

For all statistical analyses, confirm that the following items are present in the figure legend, table legend, main text, or Methods section.

| n/a | Confirmed | |
|---|---|---|
| ☐ | ☒ | The exact sample size ($n$) for each experimental group/condition, given as a discrete number and unit of measurement |
| ☐ | ☒ | A statement on whether measurements were taken from distinct samples or whether the same sample was measured repeatedly |
| ☐ | ☒ | The statistical test(s) used AND whether they are one- or two-sided *Only common tests should be described solely by name; describe more complex techniques in the Methods section.* |
| ☒ | ☐ | A description of all covariates tested |
| ☐ | ☒ | A description of any assumptions or corrections, such as tests of normality and adjustment for multiple comparisons |
| ☐ | ☒ | A full description of the statistical parameters including central tendency (e.g. means) or other basic estimates (e.g. regression coefficient) AND variation (e.g. standard deviation) or associated estimates of uncertainty (e.g. confidence intervals) |
| ☐ | ☒ | For null hypothesis testing, the test statistic (e.g. $F$, $t$, $r$) with confidence intervals, effect sizes, degrees of freedom and $P$ value noted *Give P values as exact values whenever suitable.* |
| ☒ | ☐ | For Bayesian analysis, information on the choice of priors and Markov chain Monte Carlo settings |
| ☒ | ☐ | For hierarchical and complex designs, identification of the appropriate level for tests and full reporting of outcomes |
| ☒ | ☐ | Estimates of effect sizes (e.g. Cohen's $d$, Pearson's $r$), indicating how they were calculated |

*Our web collection on statistics for biologists contains articles on many of the points above.*

## Software and code

Policy information about availability of computer code

| Data collection | Electron microscopy data were collected using SerialEM 4.1.0, analytical chromatography data were collected using UNICORN 7, SEC-MALS data were collected using ASTRA V, and ATPase and mtMDH refolding data were collected using SoftMax Pro v7. |
|---|---|
| Data analysis | Data were analyzed using MotionCor2 v1.6.4, cryoSPARC v3.3, RELION v3.1, Coot v0.8.9.2, ISOLDE v1.3, Phenix v1.20.1, UCSF Chimera v1.16, UCSF ChimeraX v1.3, MUSCLE v5, MView v1.67, Adobe Illustrator 2023, and GraphPad Prism v9.3.1. |

For manuscripts utilizing custom algorithms or software that are central to the research but not yet described in published literature, software must be made available to editors and reviewers. We strongly encourage code deposition in a community repository (e.g. GitHub). See the Nature Portfolio guidelines for submitting code & software for further information.

## Data

Policy information about availability of data

All manuscripts must include a data availability statement. This statement should provide the following information, where applicable:
- Accession codes, unique identifiers, or web links for publicly available datasets
- A description of any restrictions on data availability
- For clinical datasets or third party data, please ensure that the statement adheres to our policy

Cryo-EM densities have been deposited at the Electron Microscopy Data Bank under accession codes EMD: 29813 (mtHsp60apo consensus), EMD: 29814 (mtHsp60apo focus), EMD: 29815 (mtHsp60ATP consensus), EMD: 29816 (mtHsp60ATP focus), EMD: 29817 (mtHsp60ATP-mtHsp10 consensus), and EMD: 29818

## Human research participants

Policy information about studies involving human research participants and Sex and Gender in Research.

| Reporting on sex and gender | N/A |
| --- | --- |
| Population characteristics | N/A |
| Recruitment | N/A |
| Ethics oversight | N/A |

Note that full information on the approval of the study protocol must also be provided in the manuscript.

# Field-specific reporting

Please select the one below that is the best fit for your research. If you are not sure, read the appropriate sections before making your selection.

☒ Life sciences  ☐ Behavioural & social sciences  ☐ Ecological, evolutionary & environmental sciences

For a reference copy of the document with all sections, see nature.com/documents/nr-reporting-summary-flat.pdf

# Life sciences study design

All studies must disclose on these points even when the disclosure is negative.

| Sample size | Cryo-EM images were collected for each sample until there was a reasonable expectation of performing reconstructions at the desired resolution (~3 Angstrom, sufficient for atomic model building). In all cases the desired resolutions were reached, indicating that the sample sizes were sufficient. Biochemical experiments were performed in biological duplicate or triplicate and technical triplicate (where applicable), in keeping with standard practice in the field. Replication attempts yielded similar results in all cases, indicating that the sample sizes used were sufficient to capture experimental variability. |
| --- | --- |
| Data exclusions | Micrographs were excluded based on poor maximum estimated resolution. Particles were excluded during 2D and 3D classification in cryoSPARC and RELION based on assignment to low-resolution or otherwise artifactual classes. One biological replicate of the mtMDH folded control in the mtMDH refolding experiments showed no activity (likely due to technical error); data from this replicate were excluded from analysis. |
| Replication | Biochemical experiments were performed in biological duplicate or triplicate and and technical triplicate (where applicable). All replication attempts were successful, with the exception of the biological replicate of the mtMDH folded control in the mtMDH refolding experiments discussed above. Cryo-EM data were randomly divided into two halves that were independently refined. |
| Randomization | Cryo-EM data was randomly assigned to two half sets during refinement. Resolution estimates are based on comparisons of reconstructions from these half sets. Biochemical experiments were not randomized as no subjective assessment of data was required. |
| Blinding | Investigators were not blinded in any cryo-EM or biochemical data collection or analysis as experimental conditions needed to be known to prepare samples, acquire data, and analyze results; furthermore, no subjective assessment of data was required. |

# Reporting for specific materials, systems and methods

We require information from authors about some types of materials, experimental systems and methods used in many studies. Here, indicate whether each material, system or method listed is relevant to your study. If you are not sure if a list item applies to your research, read the appropriate section before selecting a response.

## Materials & experimental systems

| n/a | Involved in the study |
|-----|----------------------|
| ☒ | Antibodies |
| ☒ | Eukaryotic cell lines |
| ☒ | Palaeontology and archaeology |
| ☒ | Animals and other organisms |
| ☒ | Clinical data |
| ☒ | Dual use research of concern |

## Methods

| n/a | Involved in the study |
|-----|----------------------|
| ☒ | ChIP-seq |
| ☒ | Flow cytometry |
| ☒ | MRI-based neuroimaging |

