## [Peer Review File · Nature Structural & Molecular Biology]

Peer Review Information

Journal: Nature Structural and Molecular Biology

Manuscript Title: Asymmetric apical domain states of mitochondrial Hsp60 coordinate substrate engagement and chaperonin assembly

Corresponding author name(s): Dr Daniel Southworth

Reviewer Comments & Decisions:

Decision Letter, initial version:
--

25th May 2023

Dear Dr. Southworth,

Thank you for submitting your manuscript "Asymmetric apical domain states of mitochondrial Hsp60 coordinate substrate engagement and chaperonin assembly". I apologize for the delay in processing your manuscript, which resulted from difficulties in obtaining referees' reports. Nevertheless, the comments from the 2 reviewers who have evaluated your manuscript are below.

Unfortunately, after carefully considering their comments, we cannot offer to publish your manuscript in Nature Structural & Molecular Biology.

You will see that while the referees find the work of potentially interesting, they raise concerns about the strength of the novel conclusions that can be drawn at this stage and the robustness of the data, as currently presented. Based on these comments, we cannot offer to publish the manuscript in NSMB.

However, if further experimentation, analysis, and revisions allow you to address the referees concerns in full, we would be prepared to consider an appeal of our decision, on the condition that no related work is published in the interim or has been accepted in our journal. Please contact me to discuss an appeal and potential revision. Please note that, until we have the opportunity to read the revised manuscript in its entirety, we cannot promise that it will be sent back for peer review.

I am sorry we could not be more positive on this occasion. I hope that you find the referees' comments useful in deciding how best to proceed.

Sincerely,

Katarzyna Ciazynska
(she/her)
Associate Editor
Nature Structural & Molecular Biology
<https://orcid.org/0000-0002-9899-2428>

Referee expertise:

Referee #1: protein folding

Referee #2: cryo-EM, chaperones

Reviewers' Comments:

Reviewer #1:

Remarks to the Author:

In this paper the authors determine the cryo-EM structures of a mutant version of mtHSP60, which shares similarities to the better understood GroEL, in the apo, ATP-bound state, and with substrate protein (SP) or client mtMDH. These are interesting and potentially provide fodder for trying to come up with mechanism of mtHSP (WT) assisted folding (structures alone are insufficient for this purpose). I think the reports of the structures are most interesting, and are certainly worth reporting. However, the authors have taken considerable liberty to speculate on the functional aspects without experiments or justification. This has given me pause and I think the authors should cut down on the amount of speculation. The fact is that there are many contradictory opinions, often without quantification, that one can pick and choose.

I urge them to address the following questions and respond to the comments. They are presented in no particular order of importance.

(1) The second paragraph where they describe the current view of GroEL function ignores a large number of studies that show that the football is the functional state. The football forms readily when GroEL (+GroES) is challenged with SPs or clients. Under these conditions, the chaperonin machinery seems to function as a parallel processing machine. In other words, when there is a job to do (fold proteins) both the chambers spring into action. Some of the references (by no means exhaustive) are: (a) Sameshima T, Iizuka R, Ueno T, Funatsu T. (2010) Denatured proteins facilitate the formation of the football-shaped GroEL-(GroES)₂ complex. *Biochem J.* 427(2):247-54; (b) *J. Biol. Chem* 287(49):41118-25. (c) X. Ye and G Lorimer *Proc Natl Acad Sci U S A* 110 (46): E4289-97. (d) Y. Dang *PNAS* 110: E4298 (2013); (e) Koike-Takeshita A, *J. Biol. Chem.* 283(35): 23774-81.

In addition, the proposal that the multiple rounds of SP-binding and release along with a mathematical model was formulated in *PNAS* 90: 4030 (1996) – Todd. The authors should cite this in addition to reference 18.

(2) The authors made efforts to show that V72I mutant is a reasonable surrogate for the WT.

Nevertheless, it is unclear whether the interesting structures are the functional states in vivo. Maybe they could comment on this.

(3) The most interesting finding is the up down arrangement of the H/I helices in the apical domain, which they argue is needed to engage both the mtHSP10 and the client protein. This important finding appears to differ from GroEL (for instance the Sigler structures for the apo state and ATP hydrolyzed state with ES bound) as well more recent structures for the R state (pre hydrolyzed state). I think the authors should provide a comparison to provide additional structural insights.

(4) Lines 238-239 they say that (referring to 3f in the Extended data fig) that "mtHSP60 inter-ring interface is significantly reduced compared to those in analogous GroEL complexes". What aspect of the interface is reduced? The authors should quantify this? This is important because they say that this explains why mtHSP60 exists as a single ring – which they propose (almost throughout the paper) is the functional state.

(5) Just like GroEL/ES, the binding of mtHSP10 produces a large conformational change. Two questions: (i) Does mtHSP10 only engage with the up apical domain, and if so how many? If less than 7 what is the status of the remaining mtHSP10 (ES analogue)? (ii) What is the volume change of the cavity when mtHSP10 binds? In other words, how much does the cavity expand?

(6) On page 11 they show that the stability of mtHSP60 (V72I) is greatly compromised in four mutations to which the mtMDH apparently binds. From this they draw the conclusion outlined in lines 353-355. It is unclear what the role of client is in the mutant experiments, which if I understand correctly were performed in the absence of mtMDH. Besides, what is the importance of these experiments.

(7) It appears that the authors suggest that when all seven mtHSP10 are bound one obtains an expanded cavity in which the SP is encapsulated. ATP hydrolysis releases the SP (lines 372-373), whether folded or not. How does the release of mtHSP10, ADP + Phosphate occur in the absence of signaling from the trans ring? Please elaborate.

(8) The asymmetry between the various subunits is an important finding and is also known in GroEL (see Xue Fei PNAS 111: 12775 (2014)). This asymmetry has important implications in the dynamics of allosteric transitions in GroEL (PNAS 103: 18939 (2006)). Here, it appears that this asymmetry is emphasized in the context of binding of the SP and mtHSP10. Are there additional implications?

(9) Lines 408-423 comments on the 7th protomer. First, there are a bunch of speculations here. Second, presumably what constitutes the 7th is stochastic, which means the conformations are highly heterogeneous. In others, in some instances the 7th protomer maybe up and in others down. If so, what does that mean for encapsulation folding etc?

(10) Lines 443-448: The authors refer to the reconstitution of aconitase. First, the cited studies do not provide a clear explanation. Second, what is probably needed to alter the folding landscape of the SP is to interact even transiently with chaperones, which would place the protein in a different region of the energy landscape. This would in part facilitate escape from a kinetic trap if possible. In fact, there are proteins where mini-chaperones (A. R. Fersht) are sufficient.

(11) In a few places, the authors suggest that the client density is not sharp, which is excellent. In

some place they suggest that this is unexpected (line 86). However, it is generally thought the the encapsulated protein is more disordered relative to the misfolded structures in the solution. This is because of multivalent binding, which is made possible due to a stretching force. In other words, chaperonins perform work on the SP by generating forces on the order of 10 or 20 pico Newtons (see page 265 in Annual Rev Biophys Biomol Struct 30: 245 (2001)). This, generically one expects that SP-chaperonin interactions unfold (at least partially) the client proteins.

(12) The last paragraph speculates without evidence (see point number in 1 in this report). In support they cite ref 62, which shows that timing of ATP hydrolysis is linked to the changes in folding rates and more importantly yield. The link to inter-ring allostery is not made. So lines 460-462, as they admit, is pure speculation. I believe the results presented here do not give the authors license to rule out the role of the football as the functional unit, especially in view of the many references pointed out in comment 1. Please remove this, which is not needed to justify the importance of this study.

Reviewer #2:

Remarks to the Author:

- Summary of the key results

The research carried out structural study for an in vitro reconstitution of a stabilizing mutant of mtHsp60, a Group-I mitochondrial chaperonin found in humans. The experiment involved various states, namely mtHsp70 apo, mtHsp60ATP, and mtHsp60ATP-mtHsp10, and included a low-resolution client protein inside the chamber. Lastly, authors biochemically characterized substrate contact residues are important for folding activity of Hsp60.

- Originality and significance

Through the use of high-resolution CryoEM, the study was able to obtain the structure of an ATP-bound Group-I chaperonin without co-chaperonin. The research also provided stepwise structural insights into the folding procedure of the human Group-I chaperonin. Furthermore, the study focused on the classification of pseudosymmetric molecules, which allowed for the identification of asymmetric features associated with client binding.

The manuscript is well-written and follows a logical description of the structures combined with some of the biochemical experiments to support the main conclusions.

-Data & methodology

The authors' conclusion that the encapsulated density in the HSP60 chaperonin structure originates from partially folded HSP60 is not strongly supported by evidence, and may be overly speculative. Although some density was observed after 3D classification, the authors should demonstrate the validity of this density and whether it represents a meaningful client, rather than a potential artifact of image analysis.

Although the authors employed mtMDH as a substrate to assess refolding activity, they did not use it to reconstruct structural data. This creates a conceptual gap, leaving it unclear why the authors did not utilize mtMDH for structural analysis as well. Additionally, the observation of mtMDH refolding

activity by mtHsp60V72I alone does not necessarily establish a correlation between the structural observations with the ambiguous client and mtMDH functional activity.

The authors imposed C7/D7 symmetry for mtHSP60 as it is a pseudosymmetric molecule. They then expanded the particle into 7 or 2-fold axis and carried out focused classification with K=50, T=40, followed by local refinement to identify any asymmetric structural variability. Subsequently, they chose certain classes for refinement based on visual evaluation. However, it remains unclear how the expanded particle with symmetry was subjected to asymmetric refinement directly.

If particles are expanded along the symmetry axis, any genuine asymmetric characteristics should be visible across all axes. For instance, if a feature is truly asymmetric, it should repeat 7 times every 360/7 degree. The authors need to provide more details regarding their image processing methodology to clarify this.

The authors' claim regarding the structural correlation between mtHSP60 and the client protein may be questionable as they relied on a very small subset of particles, such as only 0.49% in the apical-only client. It is unclear whether such a limited population is statistically significant and representative of the overall structure.

The authors need to provide reproducible details whether the particle classification was carried out in a subjective manner by the authors. Given that they used T=40 for classification and the client protein did not have any high-resolution features, it is doubtful whether such a condition could be considered to classify the unfeatured density.

- Suggested improvements: experiments, data for possible revision

Providing a more detailed description of the authors' procedure for handling asymmetry in the pseudo-symmetric complex would be beneficial. This would help readers better understand the methodology used to analyze the structure and evaluate the reliability of the results. Additionally, including information on how the authors validated their approach would further enhance the credibility of their findings.

Referring to the GroEL/S football complex structure with the client protein (<https://doi.org/10.1016/j.isci.2021.103704>) can be a useful way to explain the conserved type I system to a broader audience.

Modify the letter in Figure1 legend (e) and (h): Colored as in (b) -> (a)
Modify the letter in Figure1 legend (i) : labeled as in (h) -> (Left)..maybe

Author Rebuttal to Initial comments

Response to Reviewers' comments

We thank the Reviewers for their time and thoughtful consideration of this manuscript. Both Reviewers commented positively about the work, indicating “*the reports of the structures are most interesting, and are certainly worth reporting*” (R1) and “*The manuscript is well-written and follows a logical description of the structures*” (R2). Additionally, the Reviewers each identify a number of important concerns and clarifications which we have specifically addressed below along with corresponding adjustments to the manuscript.

Reviewer 2 raises an important point about the identity of the central density in our structures, which we felt necessary to address with further experimentation. In short, we sought to provide further evidence that this density is an unfolded/misfolded client and thus its positioning and mtHsp60 structure in different states are directly relevant to defining the mechanism of mtHsp60-promoted client folding. To that end, we performed biochemical and structural experiments with an established client, mitochondrial malate dehydrogenase¹ (mtMDH, Fig. 5, a panel of which is reproduced here for convenience). By cryo-EM 2D analysis we identify much stronger globular density in the folding chamber of mtHsp60 following incubation with previously denatured mtMDH, compared to mtHsp60^{V72I} alone. We further identify mtHsp60 association with mtMDH by co-elution during gel filtration analysis (Fig. 5a,b). Together these observations indicate that the density we observe in the central cavity of certain classes of mtHsp60 indeed corresponds to an unfolded client, confirming the relevance of our work for understanding chaperonin folding mechanisms. Unfortunately, we were unable to obtain high-resolution reconstructions of mtMDH-bound mtHsp60 complexes due to preferred orientation issues (only top views were obtained) and/or sample dissociation, most likely due to the presence of guanidinium carried over upon addition of mtMDH. Nonetheless, our additional experiments and analysis further support our conclusions, which have resulted in an improved manuscript that provides a comprehensive structural view of human mtHsp60 function.

Additionally, during review of this manuscript a study from the group of Helen Saibil was published on *bioRxiv*, reporting client- and nucleotide-bound GroEL structures at various points in its chaperone cycle². Importantly, alternating apical domain arrangements in the absence of co-chaperonin are also observed in this study (similar but not identical to those presented in our study), underscoring the biological significance of our findings. Thus, our study establishes alternating asymmetry as a novel and conserved feature of chaperonin function that may be further defined with additional experimentation.

Fig. 5c. Selected 2D top-view class averages from the mtHsp60^{ppp} dataset (top), showing classes with no or weak client density, and from the dataset with denatured FITC-mtMDH added (bottom), showing much stronger client density. Scale bars equal 100 Å.

Responses to specific comments and concerns:

Reviewer 1:

“...the authors have taken considerable liberty to speculate on the functional aspects without experiments or justification. This has given me pause and I think the authors should cut down on the amount of speculation. The fact is that there are many contradictory opinions, often without quantification, that one can pick and choose.”

For this revision we have sought to reduce the number of discussion points that may be considered overly speculative. Importantly, R1 brings up several specific instances which we address in the below points.

(1) The second paragraph where they describe the current view of GroEL function ignores a large number of studies that show that the football is the functional state. The football forms readily when GroEL (+GroES) is challenged with SPs or clients. Under these conditions, the chaperonin machinery seems to function as a parallel processing machine. In other words, when there is a job to do (fold proteins) both the chambers spring into action. Some of the references (by no means exhaustive) are: (a) Sameshima T, Iizuka R, Ueno T, Funatsu T. (2010) Denatured proteins facilitate the formation of the football-shaped GroEL-(GroES)₂ complex. *Biochem J.* 427(2):247-54; (b) *J. Biol. Chem* 287(49):41118-25. (c) X. Ye and G Lorimer *Proc Natl Acad Sci U S A* 110 (46): E4289-97. (d) Y. Dang *PNAS* 110: E4298 (2013); (e) Koike-Takeshita A, *J. Biol. Chem.* 283(35): 23774-81.

Indeed, we appreciate the extensive body of work demonstrating the existence of symmetric complexes in the GroEL/ES cycle upon the addition of client; it was not our intention to refute the conclusions of these studies by excluding them, but rather to simplify the presentation of previous knowledge in our introduction. Upon review, we agree that including discussion of these states is relevant and would strengthen our manuscript, and have thus added and referenced such discussion (lines 54-57). We thank the Reviewer for this suggestion.

In addition, the proposal that the multiple rounds of SP-binding and release along with a mathematical model was formulated in *PNAS* 90: 4030 (1996) – Todd. The authors should cite this in addition to reference 18.

We have now included this reference – we thank the Reviewer for this suggestion.

(2) The authors made efforts to show that V72I mutant is a reasonable surrogate for the WT. Nevertheless, it is unclear whether the interesting structures are the functional states *in vivo*. Maybe they could comment on this.

While we have taken steps to demonstrate that the V72I mutant is an acceptable substitute for WT mtHsp60 (Fig. 1, SEC-MALS showing altered proportions of the same oligomeric states (heptamer vs monomer), and retention of mtMDH refolding activity, and Extended Data Fig. 1b, similar mtHsp10-dependent increase in steady-state ATPase activity), whether any mtHsp60 structure determined *in vitro* is a member of the conformational ensemble *in vivo* is at present unclear. Indeed, while general aspects of mtHsp60 mechanism determined from *in vitro* experiments mirror observations *in vivo*, for example, the dependence of mtHsp10 on folding of

a subset of clients³, further studies will be needed to more precisely define mtHsp60 structure and function *in vivo*. These studies will likely involve high-resolution crosslinking-mass spectrometry, *in vivo* FRET, or perhaps most promisingly, high-resolution cellular cryo-electron tomography, which we have now included in the Discussion as a potential means to address the issue of whether single- or double-ring complexes are present *in vivo* (lines 508-510).

(3) The most interesting finding is the up down arrangement of the H/I helices in the apical domain, which they argue is needed to engage both the mtHSP10 and the client protein. This important finding appears to differ from GroEL (for instance the Sigler structures for the apo state and ATP hydrolyzed state with ES bound) as well more recent structures for the R state (pre hydrolyzed state). I think the authors should provide a comparison to provide additional structural insights.

The differences between the apical domain conformations of the ATP-bound structure determined here and those of nucleotide-bound GroEL (particularly in the R state) are indeed interesting. Perhaps the most salient structure for comparison (and thus what we have now included) is a crystal structure of GroEL in the R-ADP state (with two mutations that favor the formation of this state)⁴. Interestingly, this structure also has markedly asymmetric apical domains, though not in the strict alternating pattern that we observe. Regardless, this structure provides additional evidence of asymmetry in the (group I) chaperonin cycle, and is in contrast to studies that assume symmetric conformations (of apical domains or otherwise), primarily using low-resolution cryo-EM reconstructions. We have included this comparison as Extended Data Fig. 5b,c and in lines 447-459, and we thank the Reviewer for this suggestion. Of note, our identification of the asymmetric states in mtHsp60 benefitted from recent advancements in cryo-EM data collection and processing, allowing for deeper characterization than previously possible. Thus, further analysis of GroEL may also reveal related conformational ensembles. Indeed, as discussed above, work by the Saibil group released on bioRxiv during the revision of this manuscript revealed similar alternating up/down apical domain states of GroEL, suggesting that this asymmetry is conserved in group I chaperonins and confirming the utility of modern cryo-EM methods to investigate chaperonin structure.

(4) Lines 238-239 they say that (referring to 3f in the Extended data fig) that “mtHSP60 inter-ring interface is significantly reduced compared to those in analogous GroEL complexes”. What aspect of the interface is reduced? The authors should quantitate this? This is important because they say that this explains why mtHSP60 exists as a single ring – which they propose (almost through out the paper) is the functional state.

The right interface of canonical chaperonin inter-ring interfaces (as now labeled in Extended Data Fig. 3f) does not feature the salt bridges found in GroEL structures. We have included a comparison of the total buried surface area of the mtHsp60^{ATP} structure with an average from three high-resolution GroEL structures (heptamer to heptamer, ~1170 Å² buried in mtHsp60 compared to ~2500 Å² in GroEL) in line 245-247. We thank the Reviewer for this suggestion.

(5) Just like GroEL/ES, the binding of mtHSP10 produces a large conformational change. Two questions: (i) Does mtHSP10 only engage with the up apical domain, and if so how many? If less than 7 what is the status of the remaining mtHSP10 (ES analogue)? (ii) What is the volume change of the cavity when mtHSP10 binds? In other words, how much does the cavity expand?

Indeed we propose that three mtHsp10 protomers initially engage with mtHsp60 based on our structures in which we identify three apical domains are in the 'up' orientation with exposed mtHsp10 binding interfaces. In this model, the mobile loops of the non-bound mtHsp10 protomers would remain unbound until a concerted structural rearrangement of the apical domains, adopting the fully extended state (the state analogous to 'R-ES' in GroEL). This rearrangement would enable the binding of the remaining mtHsp10 loops. In support of this model, a multi-step binding mechanism has been proposed for GroEL/ES primarily using biophysical data⁵⁻⁸, though to our knowledge there is no published structural information and thus direct evidence of chaperonin-co-chaperonin intermediate states that we suggest exist. Secondly, the volume of the central cavity of group I chaperonins upon co-chaperonin binding increases roughly twofold (for example, apo GroEL and the GroES-bound state are $\sim 85,000 \text{ \AA}^3$ and $\sim 175,000 \text{ \AA}^3$, respectively⁹). Given the structural similarities of the GroEL system to mtHsp60, we consider this an acceptable approximation.

(6) On page 11 they show that the stability of mtHSP60 (V72I) is greatly compromised in four mutations to which the mtMDH apparently binds. From this they draw the conclusion outlined in lines 353-355. It is unclear what the role of client is in the mutant experiments, which if I understand correctly were performed in the absence of mtMDH. Besides, what is the importance of these experiments?

We apologize for the lack of clarity – we sought to determine the importance of specific mtHsp60-client contacts by analyzing mutants in the wild-type background (not V72I), for a proper assessment of function. We have now specified in what background we made these mutations (line 341-343). We tested the ability of the mutants to refold denatured mtMDH, as well as ATPase activity as a concentration of mtHsp10, in order to understand which mtHsp60 functions were affected. Given that three of the mutants did not exhibit mtHsp10-dependent ATPase activity, we suspected and confirmed by SEC that they are deficient in oligomerization (which appears to be a general property of mtHsp60). Thus, while the data in Fig. 4 are essentially negative with respect to the analysis of the contribution of client-contacting residues to refolding activity, we feel that the striking oligomerization effects caused by single-residue mutations is interesting and will be useful knowledge in future investigations of the mtHsp60 system. Additionally, to our knowledge these mutants have not been investigated in mtHsp60 before, and few mutations in general have been reported in mtHsp60, likely due to the decreased oligomeric stability of this system. Our results therefore substantively add to the body of work on mtHsp60.

(7) It appears that the authors suggest that when all seven mtHSP10 are bound one obtains an expanded cavity in which the SP is encapsulated. ATP hydrolysis releases the SP (lines 372-373), whether folded or not. How does the release of mtHSP10, ADP + Phosphate occur in the absence of signaling from the trans ring? Please elaborate.

In the mitochondrial system the affinity of chaperonin for co-chaperonin in the presence of ADP is significantly reduced relative to GroEL/ES¹⁰. Thus, after hydrolysis, ADP, phosphate, and co-chaperonin release likely occurs immediately and is not dependent on the allosteric signal of nucleotide binding to the trans ring, enabling substrate to diffuse out of the cavity. Single mtHsp60 rings are therefore able to release substrate. We have added this explanation to the referenced section.

(8) The asymmetry between the various subunits is an important finding and is also known in GroEL (see Xue Fei PNAS 111: 12775 (2014)). This asymmetry has important implication in the dynamics of allosteric transitions in GroEL (PNAS 103: 18939 (2006)). Here, it appears that this asymmetry is emphasized in the context of binding of the SP and mtHSP10. Are there additional implications?

We thank the Reviewer for pointing out the importance of the subunit asymmetry we identify. Indeed, while we initially considered the implications of the asymmetric apical domain conformations primarily in the ATP-bound state (for binding of client and co-chaperonin), we have added further discussion and accompanying citations suggested by the Reviewer about this state and the transition to the mtHsp10-bound state (lines 460-468). Specifically, as discussed in the first reference above, apical domain asymmetry in the apo state may present distinct client-binding surfaces, which may partially explain the broad client selectivity exhibited by group I chaperonins. Additionally, our results raise interesting energetic hypotheses about mtHsp10 binding to the ATP-bound state: initial association of 3 mtHsp10 molecules would likely retain a significant degree of mtHsp10 flexibility (of the 4 unbound protomers) and thus may be entropically favorable. The existence of asymmetric intermediates (here and at other points in the cycle) are supported by the second reference above, which computationally demonstrates that there are many possible pathways for chaperonin complexes to assume the well-known (presumably stable/long-lived) structurally characterized states.

(9) Lines 408-423 comments on the 7th protomer. First, there are bunch of speculations here. Second, presumably what constitutes the 7th is stochastic, which means the conformations are highly heterogeneous. In others, in some instances the 7th protomer maybe up and in others down. If so, what does that mean for encapsulation folding etc?

We agree that the most likely explanation for the weakly resolved 7th protomer is pronounced conformational heterogeneity – we have reworked this paragraph to emphasize that point and removed additional points that may be considered too speculative. As now included, we postulate that the conformation of the 7th protomer may not contribute to chaperone function, given the six other highly ordered apical domains that we suggest are linked to co-chaperonin recruitment and client binding – future work may address this hypothesis.

(10) Lines 443-448: The authors refer to the reconstitution of aconitase. First, the cited studies do not provide a clear explanation. Second, what is probably needed to alter the folding landscape of the SP is interact even transiently with chaperones, which would place the protein in a different region of the energy landscape. This would in part facilitate escape from a kinetic trap is possible. In fact, there are proteins where mini-chaperones (A. R. Fersht) are sufficient. We thank the Reviewer for pointing this out. We agree that these studies do not provide a clear explanation about the relevance of the up/down apical states that we observe on mechanisms of folding without encapsulation. We have simplified the Discussion, as suggested by the Reviewer, and removed these points, which, as the Reviewer notes, are unable to be experimentally supported at present.

(11) In a few places, the authors suggest that the client density is not sharp, which is excellent. In some place they suggest that this is unexpected (line 86). However, it is generally thought the

the encapsulated protein is more disordered relative to the misfolded structures in the solution. This is because of multivalent binding, which is made possible due to a stretching force. In other words, chaperonins perform work on the SP by generating forces on the order of 10 or 20 pico Newtons (see page 265 in Annual Rev Biophys Biomol Struct 30: 245 (2001)). This, generically one expects that SP-chaperonin interactions unfold (at least partially) the client proteins.

We apologize for the confusion – we meant that the presence of any client was surprising (as we did not add exogenous material), not that it is surprising that the client density is low resolution. We have reworded this sentence so as to resolve the ambiguity. We agree that the client density is expected to be disordered and unresolvable given the multivalent interactions and partially folded client states resulting in heterogeneity between particles, we thank the Reviewer for the additional insight.

(12) The last paragraph speculates without evidence (see point number in 1 in this report). In support they cite ref 62, which shows that timing of ATP hydrolysis is linked to the changes in folding rates and more importantly yield. The link to inter-ring allostery is not made. So lines 460-462, as they admit, is pure speculation. I believe the results presented here do not give the authors license to rule out the role of the football as the functional unit, especially in view of the many references pointed out in comment 1. Please remove this, which is not needed to justify the importance of this study.

We agree that the speculation here is not necessary or helpful, and we have removed it – we thank the Reviewer for this suggestion. Indeed, given that (symmetric) double-ring structures have been observed in both mtHsp60 and GroEL, these states may define the functionally relevant forms of chaperonins generally.

Reviewer 2:

The authors' conclusion that the encapsulated density in the HSP60 chaperonin structure originates from partially folded HSP60 is not strongly supported by evidence, and may be overly speculative. Although some density was observed after 3D classification, the authors should demonstrate the validity of this density and whether it represents a meaningful client, rather than a potential artifact of image analysis.

The Reviewer brings up an important point about the validity of the proposed client density encapsulated by Hsp60 we observe in our 2D and 3D classification analysis. As discussed, we postulate that this density corresponds to mtHsp60 monomer carried through from our protein preparations considering that Hsp60 is a known client of itself¹¹ and the absence of other residual proteins in the prep. However, further validation of this specific density is a challenge. Thus, as discussed above, to further strengthen our work, we have now performed biochemical and structural analysis of the interaction of mtHsp60 with an established client, mitochondrial malate dehydrogenase (mtMDH), the results of which are presented in what is now Fig. 5 and associated text (lines 368-395). We find that mtMDH only interacts with mtHsp60 heptamers when it has been denatured, thus confirming the expected behavior of a typical chaperonin client¹². Following incubation of mtHsp60^{apo} heptamers with denatured mtMDH, we find an increase in top-view 2D class averages exhibiting strong central density compared to averages from the analogous dataset without added client. From these data we conclude that unfolded mtMDH is retained by Hsp60 and corresponds to an increased occupancy of client density in mtHsp60 chamber observed by cryo-EM. Given the similar appearance of densities with and without added mtMDH, we consider the conclusions drawn from our analysis of client interactions in all mtHsp60^{V72I} samples to be valid and representative of authentic states in the chaperone cycle. Unfortunately, high-resolution 3D structures of mtHsp60 incubated with mtMDH were not resolvable due to instability of the complex. While we are further pursuing structural work of mtHsp60 with mtMDH, we postulate this instability is due to the low-level presence of GdnHCl carried over from denaturation of mtMDH and thus will require substantial optimization. Additionally, these orthogonal data help rule out the potential for artifacts in image analysis, as suggested by the Reviewer. We also note that the potential for the central density to be artifactual is additionally unlikely because the 2D averages and initial models were generated by reference-free methods without any symmetry imposed. Indeed, artifactual central density can sometimes occur in structures with high-order imposed symmetry and through over-refinement. However, the asymmetric nature of the density we observe and our asymmetric refinements (see Methods) exclude this possibility. Overall, we feel our new analysis of mtHsp60 binding to mtMDH sufficiently supports our structures and conclusions that the up/down apical domain states density as bona fide client-bound states on-path to Hsp10 binding and client refolding.

Although the authors employed mtMDH as a substrate to assess refolding activity, they did not use it to reconstruct structural data. This creates a conceptual gap, leaving it unclear why the authors did not utilize mtMDH for structural analysis as well. Additionally, the observation of mtMDH refolding activity by mtHsp60^{V72I} alone does not necessarily establish a correlation between the structural observations with the ambiguous client and mtMDH functional activity. We agree with the Reviewer that further structural/functional characterization of mtMDH as an Hsp60 client is important. As discussed in the previous response, we have now performed

analyses with mtMDH including 2D classification that confirms the presence of additional client in the central chamber of Hsp60. Unfortunately, we were unable to obtain 3D reconstructions of the mtHsp60^{apo}-denatured FITC-mtMDH sample due to sample instability. Additionally, there was a severe top-view orientation bias that could not readily be overcome. While we are continuing test conditions for structural characterization, we note that substantial efforts were made for resolving mtHsp60:mtMDH at high resolution including tilted datasets, different vitrification additives and nucleotide conditions. Of note, these efforts underscore the challenges associated with studying mtHsp60 as compared to other homologs (also shown in Fig. 1), rationalizing our choice of material and associated analysis. However, our additional analyses and previous mtHsp60-promoted mtMDH refolding data, support our overall conclusions.

The authors imposed C7/D7 symmetry for mtHSP60 as it is a pseudosymmetric molecule. They then expanded the particle into 7 or 2-fold axis and carried out focused classification with K=50, T=40, followed by local refinement to identify any asymmetric structural variability. Subsequently, they chose certain classes for refinement based on visual evaluation. However, it remains unclear how the expanded particle with symmetry was subjected to asymmetric refinement directly.

We apologize for the lack of clarity. To summarize the steps of our workflow, for all three states we indeed performed refinements with symmetry imposed (C7 (apo) or D7 (ATP- and mtHsp10-bound)). Then, we symmetry-expanded the particles in C7 (apo), D7 (ATP-bound), or C2 (mtHsp10-bound) and performed focused classification without image alignment, resulting in the shown asymmetric classes of interest (Extended Data Figs. 1d, 3b, 4b). For each desired class, we then performed a final local refinement in cryoSPARC without symmetry imposed (and default parameters), such that the approximate poses of the symmetry-expanded particles were maintained (i.e. particles did not jump to another pseudo-symmetric position). Importantly, cryoSPARC requires the use of a mask for local refinement, so masks were created to encompass the entire ring in question for each state, excluding density from the other ring (in ATP- and mtHsp10-bound states). No further refinements were performed after this step. We have updated the Methods to more clearly describe this process (as an example, apo state, lines 778-809).

If particles are expanded along the symmetry axis, any genuine asymmetric characteristics should be visible across all axes. For instance, if a feature is truly asymmetric, it should repeat 7 times every 360/7 degree. The authors need to provide more details regarding their image processing methodology to clarify this.

Indeed, one should ideally expect features from a particle stack that has been symmetry-expanded to appear at every symmetry-related orientation. However, there are several reasons why we do not believe the lack of all possible orientations in the resulting classes is problematic. Using the mtHsp60^{apo} dataset as an initial example, the apical domains can be considered to exhibit continuous conformational variability, rendering the number of unique heptameric apical domain configurations far too large to successfully classify using standard methods. We sorted the data into a significant number of classes (50, which is much larger than the number used in standard workflows) in order to sufficiently recover many high-quality maps (either with high-resolution apical domains or with strong client density), but there are simply not enough classes to obtain “correct” reconstructions for every unique state at every possible orientation. Thus, we

consider the high-quality output classes we identified to be an incomplete but representative sampling of the true structural heterogeneity in the dataset. Notably, in our mtHsp60^{ATP} processing we observe identical states at four of seven symmetry-related positions (Extended Data Fig. 3b), indicating that classification in this manner is able to partially satisfy the expectation of seven such classes. However, the four identical reconstructions observed in this classification were likely due to the significantly reduced conformational variability of the apical domain in the ATP state (essentially two ordered states, compared to the continuous variability in mtHsp60^{apo}). Even still, seven copies are presumably not observed due to the number of classes used (50 again), which is apparently insufficient to describe all heterogeneity in the dataset but was necessary for the execution of this job given the computational resources available to us. We have added more details in the Methods about our classification methodology that clarify how the final asymmetric maps were obtained (as an example, apo state, lines 778-809).

The authors' claim regarding the structural correlation between mtHSP60 and the client protein may be questionable as they relied on a very small subset of particles, such as only 0.49% in the apical-only client. It is unclear whether such a limited population is statistically significant and representative of the overall structure.

As discussed above, we used a large number of classes (50) to separate mtHsp60^{apo} particles on the basis of apical domain and client conformation. Though we discuss only four classes in the main text (one with high-resolution apical domains and three with client at distinct positions), there were additional classes with similar features that support the significance of the four we chose. To provide a more complete description of our results, we have included summary statistics for the focused classification job in question (Extended Data Fig. 1g) which show that most classes (36/50) correspond to complexes with asymmetric apical domains, and 13/50 correspond to those with client density (also with asymmetric apical domains). Thus, while the class we show indeed represents a small proportion of the total particle number, there are many such classes that constitute an appreciable fraction of the data. We have included additional discussion on this point in lines 180-184 of the Results section, and we thank the Reviewer for bringing up this point.

The authors need to provide reproducible details whether the particle classification was carried out in a subjective manner by the authors. Given that they used T=40 for classification and the client protein did not have any high-resolution features, it is doubtful whether such a condition could be considered to classify the unfeatured density.

Indeed, finding conditions (in effect, mask geometry, number of classes K, and regularization parameter T) to accurately classify the client in all states was a major part of this work. We absolutely agree that tightly masking around the client would lead to overfitting, as there is very little ordered density (perhaps none) that would repeat across many particles and thus lead to a quality reconstruction. Therefore, in each dataset we carefully optimized the size of the mask used for focused classification, aiming to include just enough mtHsp60 density (namely, the apical domains) to obtain high-resolution reconstructions while enhancing the contribution of client density to the classification (by masking out some mtHsp60 density). In this manner we hypothesized that client density may be somewhat correlated to mtHsp60 conformation, which we consider correct based on our results, especially in the ATP-bound state (though not in the

case of the mtHsp10-bound complex, where mtHsp60 does not display any notable conformational heterogeneity). In addition to the masks used for focused classification already shown (Extended Data Figs. 1d, 3b), we have added the mask used for the analogous job in the football complex (Extended Data Fig. 4b), as well as discussed the optimization of the job parameters in the Methods section (lines 793-798). Of note, the high T value employed during classification produced better results (both by resolution and general map quality), likely due to the significant heterogeneity in these datasets.

Providing a more detailed description of the authors' procedure for handling asymmetry in the pseudo-symmetric complex would be beneficial. This would help readers better understand the methodology used to analyze the structure and evaluate the reliability of the results. Additionally, including information on how the authors validated their approach would further enhance the credibility of their findings.

As discussed above, we have in the Methods section included additional details about the focused classification procedures used.

Referring to the GroEL/S football complex structure with the client protein (<https://doi.org/10.1016/j.isci.2021.103704>) can be a useful way to explain the conserved type I system to a broader audience.

We thank the Reviewer for this suggestion, which was also raised by Reviewer 1 (see above, point 1). We have added discussion of the GroEL/ES football to the introduction (lines 54-57) and discussion (lines 506-508).

Reviewer 2 additional points:

Modify the letter in Figure1 legend (e) and (h): Colored as in (b) -> (a)
Agreed and corrected.

Modify the letter in Figure1 legend (i) : labeled as in (h) -> (Left)..maybe
Agreed and corrected.

References

1. Bie, A. S. *et al.* An inventory of interactors of the human HSP60/HSP10 chaperonin in the mitochondrial matrix space. *Cell Stress Chaperones* (2020) doi:10.1007/s12192-020-01080-6.
2. Gardner, S., Darrow, M. C., Lukyanova, N., Thalassinou, K. & Saibil, H. R. Structural basis of substrate progression through the chaperonin cycle. *bioRxiv* (2023) doi:10.1101/2023.05.29.542693.
3. Höhfeld, J. & Hartl, F. U. Role of the chaperonin cofactor Hsp10 in protein folding and sorting in yeast mitochondria. *J. cell Biol.* **126**, 305–315 (1994).

4. Fei, X., Yang, D., LaRonde-LeBlanc, N. & Lorimer, G. H. Crystal structure of a GroEL-ADP complex in the relaxed allosteric state at 2.7 Å resolution. *Proc National Acad Sci* **110**, E2958–E2966 (2013).
5. Cliff, M. J., Limpkin, C., Cameron, A., Burston, S. G. & Clarke, A. R. Elucidation of Steps in the Capture of a Protein Substrate for Efficient Encapsulation by GroE. *J Biol Chem* **281**, 21266–21275 (2006).
6. Miyazaki, T. *et al.* GroEL-Substrate-GroES Ternary Complexes Are an Important Transient Intermediate of the Chaperonin Cycle*. *J Biol Chem* **277**, 50621–50628 (2002).
7. Taniguchi, M., Yoshimi, T., Hongo, K., Mizobata, T. & Kawata, Y. Stopped-flow Fluorescence Analysis of the Conformational Changes in the GroEL Apical Domain RELATIONSHIPS BETWEEN MOVEMENTS IN THE APICAL DOMAIN AND THE QUATERNARY STRUCTURE OF GroEL*. *J Biol Chem* **279**, 16368–16376 (2004).
8. Kawata, Y. *et al.* Functional Communications between the Apical and Equatorial Domains of GroEL through the Intermediate Domain †. *Biochemistry-us* **38**, 15731–15740 (1999).
9. Xu, Z., Horwich, A. L. & Sigler, P. B. The crystal structure of the asymmetric GroEL–GroES–(ADP)₇ chaperonin complex. *Nature* **388**, 741–750 (1997).
10. Nielsen, K. L. & Cowan, N. J. A Single Ring Is Sufficient for Productive Chaperonin-Mediated Folding In Vivo. *Mol Cell* **2**, 93–99 (1998).
11. Cheng, M. Y., Hartl, F.-U. & Horwich, A. L. The mitochondrial chaperonin hsp60 is required for its own assembly. *Nature* **348**, 455–458 (1990).
12. Viitanen, P. V., Gatenby, A. A. & Lorimer, G. H. Purified chaperonin 60 (groEL) interacts with the nonnative states of a multitude of Escherichia coli proteins. *Protein Sci.* **1**, 363–369 (1992).

Decision Letter, first revision:

15th Jan 2024

Dear Dr. Southworth,

Thank you again for submitting your manuscript "Asymmetric apical domain states of mitochondrial Hsp60 coordinate substrate engagement and chaperonin assembly". [I apologize for the delay in responding, which resulted from the difficulty in obtaining suitable referee reports. Nevertheless, we now have comments (below) from the 2 reviewers who evaluated your paper. In light of those reports, we remain interested in your study and would like to see your response to the comments of the referees, in the form of a revised manuscript.

You will see that while reviewer #1 had no further comments, reviewer #2 has remaining concerns regarding the data processing approach. We ask you to make a final effort in addressing these.

Please be sure to address/respond to all concerns of the referees in full in a point-by-point response and highlight all changes in the revised manuscript text file. If you have comments that are intended for editors only, please include those in a separate cover letter.

We expect to see your revised manuscript within 6 weeks. If you cannot send it within this time, please contact us to discuss an extension; we would still consider your revision, provided that no similar work has been accepted for publication at NSMB or published elsewhere.

Reporting Summary:

When submitting the revised version of your manuscript, please pay close attention to our [href="https://www.nature.com/nature-portfolio/editorial-policies/image-integrity">Digital Image Integrity Guidelines](https://www.nature.com/nature-portfolio/editorial-policies/image-integrity). and to the following points below:

Finally, please ensure that you retain unprocessed data and metadata files after publication, ideally archiving data in perpetuity, as these may be requested during the peer review and production

process or after publication if any issues arise.

Please note that all key data shown in the main figures as cropped gels or blots should be presented in uncropped form, with molecular weight markers. These data can be aggregated into a single supplementary figure item. While these data can be displayed in a relatively informal style, they must refer back to the relevant figures. These data should be submitted with the final revision, as source data, prior to acceptance, but you may want to start putting it together at this point.

Data availability: this journal strongly supports public availability of data. All data used in accepted papers should be available via a public data repository, or alternatively, as Supplementary Information. If data can only be shared on request, please explain why in your Data Availability Statement, and also in the correspondence with your editor. Please note that for some data types, deposition in a public repository is mandatory - more information on our data deposition policies and available repositories can be found below:
<https://www.nature.com/nature-research/editorial-policies/reporting-standards#availability-of-data>

Nature Structural & Molecular Biology is committed to improving transparency in authorship. As part of our efforts in this direction, we are now requesting that all authors identified as 'corresponding author' on published papers create and link their Open Researcher and Contributor Identifier (ORCID) with their account on the Manuscript Tracking System (MTS), prior to acceptance. This applies to primary research papers only. ORCID helps the scientific community achieve unambiguous attribution of all scholarly contributions. You can create and link your ORCID from the home page of the MTS by

clicking on 'Modify my Springer Nature account'. For more information please visit please visit www.springernature.com/orcid.

[Redacted]

Sincerely,

Katarzyna Ciazynska, PhD
(she/her)
Associate Editor
Nature Structural & Molecular Biology
<https://orcid.org/0000-0002-9899-2428>

Reviewers' Comments:

Reviewer #1:

Remarks to the Author:

The authors have done a great job of addressing my concerns. I recommend publication.

Reviewer #2:

Remarks to the Author:

The authors supplied further details regarding their image processing. However, the rationale behind addressing pseudosymmetry appears unclear and biased. As highlighted by the authors, pseudosymmetry introduces extreme heterogeneity beyond the capacity of existing programs. For instance, the authors noted their inability to detect all rotational repeats of asymmetric features along every expanded axis. This suggests that particles are somehow mixed in pseudosymmetry, and the classification strategy may not be optimal. In such a scenario, rather than relying on symmetry imposition and particle expansion, the authors should contemplate non-symmetric analysis, as demonstrated in the work of Helen Sabil and other studies.

The author did not specify the methodology employed for eliminating duplicated particles following the 3D classification. To illustrate, in Extended Data Figure 1, Classes 1, 2, 3, and 4 (comprising 839,799 particles) underwent expansion to 'x' particles, assuming a count of 5.8K. Subsequently, four classes were selected from a pool of 50 classes, and the 'x' particles were re-extracted. These re-extracted

particles were then subjected to local refinement. To ensure clarity, the author should detail the process of confirming that the final refinement does not include duplicated particles, especially given the original expansion of 7 times.

Author Rebuttal, first revision:

Response to Reviewers' comments

We thank the Reviewers for their further consideration of this manuscript. Below, we address outstanding concerns about the methodology employed for cryo-EM structure determination.

Responses to specific comments and concerns:

Reviewer 2:

The authors supplied further details regarding their image processing. However, the rationale behind addressing pseudosymmetry appears unclear and biased. As highlighted by the authors, pseudosymmetry introduces extreme heterogeneity beyond the capacity of existing programs. For instance, the authors noted their inability to detect all rotational repeats of asymmetric features along every expanded axis. This suggests that particles are somehow mixed in pseudosymmetry, and the classification strategy may not be optimal. In such a scenario, rather than relying on symmetry imposition and particle expansion, the authors should contemplate non-symmetric analysis, as demonstrated in the work of Helen [Saibil] and other studies.

The Reviewer raises an important point about the results of image processing with or without symmetry imposition and symmetry expansion. In short, we employed symmetry expansion to allow particles in distinct symmetry-related orientations to be classified together without additional image alignment, making these jobs computationally tractable. We argue that this approach generates higher quality classes than those obtained from non-symmetric analysis, and does not introduce symmetry-related artifacts. We here substantiate this claim by comparing our original maps (using symmetry-expanded particles) from classification to those obtained without symmetry expansion, focusing on the ATP state (Fig. 1 for Reviewers). We investigated 1) whether individual pairs of apical domains are observed to adopt the up/down conformation using non-symmetry-expanded particles and 2) whether complete apical domain rings exist in similar conformations as observed with symmetry expansion. To that end, we performed focused classification on particles from a C1 refinement, without image alignment. We used two masks, one encompassing two adjacent apical domains, and the other encompassing all seven apical domains in a single ring (the same mask used in the job presented in the manuscript). Using the mask with two apical domains, we identify two classes with both apical domains resolved, in up/down (class 5) and down/up (class 8) conformations, respectively (Fig. 1a for Reviewers). Rigid-body docking appropriate pairs of apical domains from the mtHsp60^{ATP} focus structure into these maps reveals an

excellent fit (Fig. 1b for Reviewers), indicating that the up/down conformations obtained with or without symmetry expansion are identical. Interestingly, we also identify two classes with one ordered apical domain and one disordered (classes 9, 10); these may correspond to the positions that include the disordered apical domain in the mtHsp60^{ATP} focus ring (see for example Fig. 2d,f in main text), though, given the resolution obtained with this approach, a definitive conclusion is not possible.

The classification using the seven apical domain mask and non-symmetry-expanded particles revealed similar classes to those presented in the manuscript, albeit with markedly reduced quality (Fig. 1a for Reviewers). Importantly, we identify a class with six ordered apical domains in alternating down and up conformations (class 10), as well as other classes with smaller sets of ordered, alternating domains (classes 1, 2, 3, 4, 6, 7, 8). Docking sets of apical domains from mtHsp60^{ATP} focus into well-resolved map regions again reveals excellent fits (Fig. 1c for Reviewers), indicating that the conformations identified using particles with or without symmetry expansion are identical. Critically, we note that the quality of maps obtained with symmetry expansion is of much higher quality relative to those without symmetry expansion presented here (Fig. 1d for Reviewers). This reduced quality is perhaps expected due to the inability of particles in different symmetry-related orientations to be combined (without image alignment). These results thus justify our use of symmetry expansion to better resolve the conformational variability in this dataset, as compared with asymmetric approaches employed by the Saibil group¹ and others. Taken together, these results indicate that our symmetry expansion approach improves classification performance and does not introduce artifacts related to the use of symmetry. Finally, we note that, independent of classification method, no adjacent up/up or down/down pairs of apical domains are observed in our ATP dataset, indicating that symmetric pairs are incompatible in the ATP state. This finding reinforces our conclusion that the novel up/down arrangement we identify in this study is the most stable and resolvable configuration in this state.

Fig. 1 for Reviewers. Focused classification of mtHsp60^{ATP} without symmetry expansion.

(a) Processing workflow for asymmetric analysis of mtHsp60^{ATP} particles. Masks used are shown as transparent surfaces overlaid with the unsharpened consensus refinement. Apical domains in the ‘down’ conformation are colored in red, and those in the ‘up’ conformation are colored in green. In both jobs, apical domains unable to be definitively identified are colored in gray. (b) Enlarged views of classes from

the 2-apical job with two ordered apical domains (transparent, colored as in **a**), with apical domains from mtHsp60^{ATP} focus (PDB: 8G7M) rigid-body docked and shown in ribbon representation. Note excellent map/model fit. **(c)** As in **(b)**, but for classes or ordered regions thereof from the 7-apical job. **(d)** Comparison of class with 6 ordered apical domains obtained without symmetry expansion (left) to one obtained with symmetry expansion (right). Note the higher quality of the map using symmetry-expanded particles.

The author did not specify the methodology employed for eliminating duplicated particles following the 3D classification. To illustrate, in Extended Data Figure 1, Classes 1, 2, 3, and 4 (comprising 839,799 particles) underwent expansion to 'x' particles, assuming a count of 5.8K. Subsequently, four classes were selected from a pool of 50 classes, and the 'x' particles were re-extracted. These re-extracted particles were then subjected to local refinement. To ensure clarity, the author should detail the process of confirming that the final refinement does not include duplicated particles, especially given the original expansion of 7 times.

The Reviewer raises an important point – it would indeed be inappropriate to have duplicate particles in the same orientation contributing to a reconstruction. However, in the symmetry expansion method, it is not necessary to remove duplicate particles because the same particle can contribute to the reconstruction at multiple symmetry-related orientations (indeed, this is the point of symmetry expansion). Thus, if one ensures that the orientations of duplicate particles do not shift to those of other copies, a reconstruction using duplicate particles is valid. We therefore did not remove duplicates from our final particle stacks, as we ensured that no shifts to other symmetry-related orientations occurred by keeping the alignment parameters used for focused classification after particle re-extraction, and then performing local refinement, such that only minor changes to these parameters are permitted. Importantly, we note that this procedure is standard practice in the field^{2,3}.

As evidence that our maps do not suffer from inappropriate particle duplication, we note that the FSC curves of our reconstructions go to 0 rather than plateauing at a higher value (a telltale sign of duplicate particles in the same orientation) (Extended Data Fig. 2 and Fig. 2 for Reviewers, left). Furthermore, we demonstrate that duplicate removal in the mtHsp60^{ATP} focus map does not significantly alter map quality or resolution estimates, though the unmasked FSC deteriorates (as a function of resolution) somewhat more rapidly in the map with duplicate removal (Fig. 2 for Reviewers). These results indicate that the duplicate particles are not the reason for the high map quality we observe in the final local refinements, and we conclude that the maps in this study are the result of valid image processing workflows.

Fig. 2 for Reviewers. mtHsp60^{ATP} focus maps with and without duplicate removal.

Unsharpened maps and corresponding FSC curves from refinements of the mtHsp60^{ATP} focus particle stack, without prior duplicate removal (left, EMD-29816), or with duplicate removal performed in cryoSPARC (right). Note similar map quality and resolution estimates. The ring used for classification and refinement (that is, contained in the mask) is indicated in the side views (*).

References

1. Gardner, S., Darrow, M. C., Lukoyanova, N., Thalassinou, K. & Saibil, H. R. Structural basis of substrate progression through the bacterial chaperonin cycle. *Proc. Natl. Acad. Sci.* **120**, (2023).
2. Scheres, S. H. W. Processing of Structurally Heterogeneous Cryo-EM Data in RELION. in vol. 579 125–157 (Elsevier, 2016).
3. Zhou, M. *et al.* Atomic structure of the apoptosome: mechanism of cytochrome c- and dATP-mediated activation of Apaf-1. *Gene Dev* **29**, 2349–2361 (2015).

Decision Letter, second revision:

Our ref: NSMB-A47493B

22nd Feb 2024

Dear Dr. Southworth,

Thank you for submitting your revised manuscript "Asymmetric apical domain states of mitochondrial Hsp60 coordinate substrate engagement and chaperonin assembly" (NSMB-A47493B). It has now been seen by the original referees and their comments are below. As you know, in the face of reviewer #2 disagreeing with the approach taken in processing the cryo-EM data, we sought arbitration, and to this end recruited referee #3 to assess the technical aspects of the study. Since the referee agreed that the employed cryo-EM processing strategy was valid, we will therefore be happy in principle to publish it in Nature Structural & Molecular Biology, pending minor revisions to satisfy the referees' final requests and to comply with our editorial and formatting guidelines.

We are now performing detailed checks on your paper and will send you a checklist detailing our editorial and formatting requirements in about 2 weeks. Please do not upload the final materials and make any revisions until you receive this additional information from us.

Sincerely,

Katarzyna Ciazynska, PhD
(she/her)
Associate Editor
Nature Structural & Molecular Biology
<https://orcid.org/0000-0002-9899-2428>

Reviewer #2 (Remarks to the Author):

I acknowledged the author's diligent efforts and conducted a thorough assessment of the response. I concur that the up/down conformations, whether obtained with or without symmetry expansion, are conceptually identical from a biological standpoint.

However, I respectfully disagree with the authors' assertion that "the same particle can contribute to the reconstruction at multiple symmetry-related orientations (indeed, this is the point of symmetry expansion)." In my perspective, this statement holds true only when considering the individual

asymmetric unit, such as an individual subunit, rather than the rings of the complex as a whole. As evidenced by the references provided by the author themselves, the standard practice in the field is to focus on the asymmetric unit after symmetry expansion.

Reviewer #3 (Remarks to the Author):

The authors have used a symmetry expansion approach to investigate the distinct conformations of the apical domains of the Hsp60 protomers, a method that is well-established in the field and appropriate for this system. The authors support their results in the reviewer response by performing additional analyses that includes both focused classification without image alignment of the entire apical ring, as well as classification of dimers. Both of these approaches are sound in theory and well-implemented, and distinct conformations are observed that are consistent between the two approaches. More importantly, they are consistent with the results of the initial symmetry-expansion approach, albeit at slightly lower resolution. Based on my expertise in cryo-EM image analysis I support the authors' approach and conclusions, and see no reason why this manuscript shouldn't proceed for publication.

Decision Letter, final checks:

Our ref: NSMB-A47493B

28th Mar 2024

Dear Dr. Southworth,

Thank you for your patience as we've prepared the guidelines for final submission of your Nature Structural & Molecular Biology manuscript, "Asymmetric apical domain states of mitochondrial Hsp60 coordinate substrate engagement and chaperonin assembly" (NSMB-A47493B). Please carefully follow the step-by-step instructions provided in the attached file, and add a response in each row of the table to indicate the changes that you have made. Ensuring that each point is addressed will help to ensure that your revised manuscript can be swiftly handed over to our production team.

In recognition of the time and expertise our reviewers provide to Nature Structural & Molecular Biology's editorial process, we would like to formally acknowledge their contribution to the external peer review of your manuscript entitled "Asymmetric apical domain states of mitochondrial Hsp60 coordinate substrate engagement and chaperonin assembly". For those reviewers who give their assent, we will be publishing their names alongside the published article.

Nature Structural & Molecular Biology offers a Transparent Peer Review option for new original research manuscripts submitted after December 1st, 2019. As part of this initiative, we encourage our authors to support increased transparency into the peer review process by agreeing to have the reviewer comments, author rebuttal letters, and editorial decision letters published as a Supplementary item. When you submit your final files please clearly state in your cover letter whether or not you would like to participate in this initiative. Please note that failure to state your preference will result in delays in accepting your manuscript for publication.

Cover suggestions

COVER ARTWORK: We welcome submissions of artwork for consideration for our cover. For more information, please see our guide for cover artwork.

Nature Structural & Molecular Biology has now transitioned to a unified Rights Collection system which will allow our Author Services team to quickly and easily collect the rights and permissions required to publish your work. Approximately 10 days after your paper is formally accepted, you will receive an email in providing you with a link to complete the grant of rights. If your paper is eligible for Open Access, our Author Services team will also be in touch regarding any additional information that may be required to arrange payment for your article.

Please note that *Nature Structural & Molecular Biology* is a Transformative Journal (TJ). Authors may publish their research with us through the traditional subscription access route or make their paper immediately open access through payment of an article-processing charge (APC). Authors will not be required to make a final decision about access to their article until it has been accepted. Find out more about Transformative Journals

Please use the following link for uploading these materials:
[Redacted]

Best regards,

Aimee Frier
Editorial Assistant
Nature Structural & Molecular Biology
nsmb@us.nature.com

On behalf of

Katarzyna Ciazynska, PhD
(she/her)
Associate Editor
Nature Structural & Molecular Biology
<https://orcid.org/0000-0002-9899-2428>

Reviewer #2:
Remarks to the Author:

I acknowledged the author's diligent efforts and conducted a thorough assessment of the response. I concur that the up/down conformations, whether obtained with or without symmetry expansion, are conceptually identical from a biological standpoint.

However, I respectfully disagree with the authors' assertion that "the same particle can contribute to the reconstruction at multiple symmetry-related orientations (indeed, this is the point of symmetry expansion)." In my perspective, this statement holds true only when considering the individual asymmetric unit, such as an individual subunit, rather than the rings of the complex as a whole. As evidenced by the references provided by the author themselves, the standard practice in the field is to focus on the asymmetric unit after symmetry expansion.

Reviewer #3:
Remarks to the Author:

The authors have used a symmetry expansion approach to investigate the distinct conformations of the apical domains of the Hsp60 protomers, a method that is well-established in the field and appropriate for this system. The authors support their results in the reviewer response by performing additional analyses that includes both focused classification without image alignment of the entire apical ring, as well as classification of dimers. Both of these approaches are sound in theory and well-

implemented, and distinct conformations are observed that are consistent between the two approaches. More importantly, they are consistent with the results of the initial symmetry-expansion approach, albeit at slightly lower resolution. Based on my expertise in cryo-EM image analysis I support the authors' approach and conclusions, and see no reason why this manuscript shouldn't proceed for publication.

Final Decision Letter:

7th Jun 2024

Dear Dr. Southworth,

We are now happy to accept your revised paper "Asymmetric apical domain states of mitochondrial Hsp60 coordinate substrate engagement and chaperonin assembly" for publication as an Article in Nature Structural & Molecular Biology.

Your paper will be published online soon after we receive proof corrections and will appear in print in the next available issue. You can find out your date of online publication by contacting the production team shortly after sending your proof corrections.

Please note that *Nature Structural & Molecular Biology* is a Transformative Journal (TJ). Authors may publish their research with us through the traditional subscription access route or make their paper immediately open access through payment of an article-processing charge (APC). Authors will not be required to make a final decision about access to their article until it has been accepted. Find out more about Transformative Journals

Sincerely,

Katarzyna Ciazynska, PhD
(she/her)
Associate Editor
Nature Structural & Molecular Biology
<https://orcid.org/0000-0002-9899-2428>
